

# Data-driven classification of the certainty of scholarly assertions

Mario Prieto[1], Helena Deus[2], Anita de Waard[3], Erik Schultes[4], Beatriz García-Jiménez[5] and Mark D. Wilkinson[1]

[1] Departamento de Biotecnología-Biología Vegetal, Escuela Técnica Superior de Ingeniería Agronómica, Alimentaria y de Biosistemas, Centro de Biotecnología y Genómica de Plantas, Universidad Politécnica de Madrid (UPM)- Instituto Nacional de Investigación y Tecnología Agraria y Alimentaria (INIA), Pozuelo de Alarcon, Madrid, Spain
[2] Elsevier Inc., Cambridge, MA, United States of America
[3] Elsevier Research Collaborations Unit, Jericho, VT, United States of America
[4] GO FAIR International Support and Coordination Office, Leiden, The Netherlands
[5] Centro de Biotecnología y Genómica de Plantas, Universidad Politécnica de Madrid (UPM)- Instituto Nacional de Investigación y Tecnología Agraria y Alimentaria (INIA), Pozuelo de Alarcon, Madrid, Spain

## ABSTRACT

The grammatical structures scholars use to express their assertions are intended to convey various degrees of certainty or speculation. Prior studies have suggested a variety of categorization systems for scholarly certainty; however, these have not been objectively tested for their validity, particularly with respect to representing the interpretation by the reader, rather than the intention of the author. In this study, we use a series of questionnaires to determine how researchers classify various scholarly assertions, using three distinct certainty classification systems. We find that there are three distinct categories of certainty along a spectrum from high to low. We show that these categories can be detected in an automated manner, using a machine learning model, with a cross-validation accuracy of 89.2% relative to an author-annotated corpus, and 82.2% accuracy against a publicly-annotated corpus. This finding provides an opportunity for contextual metadata related to certainty to be captured as a part of text-mining pipelines, which currently miss these subtle linguistic cues. We provide an exemplar machine-accessible representation—a Nanopublication—where certainty category is embedded as metadata in a formal, ontology-based manner within text-mined scholarly assertions.

## INTRODUCTION

Narrative scholarly articles continue to be the norm for communication of scientific results. While there is an increasing push from both journals and funding agencies to publish source data in public repositories, the resulting article, containing the interpretation of that data and the reasoning behind those conclusions, continues to be, by and large, textual. The norms of scholarly writing and scholarly argumentation are learned by students as they progress through their careers, with the rules of scholarly expression being enforced by journal editors and reviewers. Among the unique features of scholarly writing is the tendency for authors to use hedging—that is, to avoid stating an assertion with certainty,

Corresponding author
Mark D. Wilkinson,
markw@illuminae.com,
mark.wilkinson@upm.es

| How a claim becomes a fact |
| --- |
| *"These miRNAs neutralize p53- mediated CDK inhibition, **possibly** through direct inhibition of the expression of the tumor suppressor LATS2." (Voorhoeve et al. 2007)* |
| *"In a genetic screen, miR-372 and miR-373 **were found to** allow proliferation of primary human cells that express oncogenic RAS and active p53, **possibly** by inhibiting the tumor suppressor LATS2 (Voorhoeve et al., 2006)." (Kloosterman and Plasterk 2006)* |
| *"[On the other hand,] two miRNAs, miRNA-372 and-373, function as **potential** novel oncogenes in testicular germ cell tumors by inhibition of LATS2 expression, **which suggests that** Lats2 is an important tumor suppressor (Voorhoeve et al., 2006)." (Yabuta et al. 2007)* |
| *"Two oncogenic miRNAs, miR-372 and miR-373, **directly inhibit** the expression of Lats2, **thereby** allowing tumorigenic growth in the presence of p53 (Voorhoeve et al., 2006)." (Okada et al. 2011)* |

**Figure 1   How a claim becomes a fact.** These statements represent a series of scholarly assertions about the same biological phenomenon, revealing that the core assertion transforms from a hedging statement into statements resembling fact through several steps, but without additional evidence (*de Waard, 2012*).

but rather to use phrases that suggest that the assertion is an interpretation of experimental evidence or speculation about a state of affairs, which is essential when presenting unproven propositions with appropriate caution (*Hyland, 1996*). For example, "*These results **suggest that** the APC is constitutively associated with the cyclin D1/CDK4 complex and are consistent with a model in which the APC is responsible for cyclin D1 proteolysis in response to IR...*" (*Agami & Bernards, 2000*); or "*With the understanding that coexpression of genes **may imply** coregulation and participation in similar biological processes…*" (*Campbell et al., 2007*). As a result, biology papers contain a wide range of argumentational structures that express varying degrees of confidence or certainty. These subtle linguistic structures become problematic, however, in the context of scholarly citation. As discussed by *De Waard & Maat (2012)*, citing papers may contain reformulations of the original claims in which the degree of certainty of the original claim is modified (and usually made stronger) in the absence of additional evidence (Fig. 1; *Latour & Woolgar, 2013*). This "drift" in certainty can be very gradual over successive steps of a citation chain, but the consequences may be profound, since statements with greater certainty than the original author intended can be used as the basis for new knowledge. Although peer-review might protect the literature from such 'hedging erosion', reviewers may lack the specific domain knowledge required to know the legacy of a given scholarly claim. Even if they take the time to follow a citation, subtle differences in expressed certainty over a single step in a citation chain may not be detectable. This problem is worsened in the context of text mining algorithms that currently do not richly capture the nuances of a scholarly assertion when extracting the entity-relationships that make up the claim.

Given that the volume of literature published grows by approximately a half-million papers per year in the biomedical domain alone, text mining is becoming an increasingly important way to capture this new knowledge in a searchable and machine-accessible way. Accurate, automated knowledge capture will therefore require accurate capture of the certainty with which the claim was expressed. Moreover, there is increasing pressure to publish knowledge, *ab initio*, explicitly for machines, in particular with the adoption of the FAIR Data Principles for scholarly publishing (*Wilkinson et al., 2016*), and
several machine-accessible knowledge publication formats have recently been suggested, including NanoPublications (*Groth, Gibson & Velterop, 2010*), and Micropublications (*Clark, Ciccarese & Goble, 2014*). In order to capture the intent of the author in these machine-readable publications, it will be necessary for them to include formal machine-readable annotations of certainty.

A number of prior studies have attempted to categorize and capture the expression of scholarly certainty. These, and other certainty categorization studies, are summarized, compared and contrasted in Table 1, where the columns represent relevant study features that distinguish these various investigations, and affect the interpretation of their outcomes. For example, the use of linguistic experts, versus biomedical domain experts, will likely affect the quality of the annotations, while using explicit rule-matching/guidelines will result in strict, predetermined categorizations. Similarly, the use of abstracts consisting of concise reporting language, versus full text which contains more exploratory narratives, will affect the kinds of statements in the corpus (*Lorés, 2004*), and their degree of certainty.

According to *Wilbur, Rzhetsky & Shatkay (2006)* "each [statement] fragment conveys a degree of certainty about the validity of the assertion it makes". While intuitively correct, it is not clear if certainty can be measured/quantified, if these quantities can be categorized or if they are more continuous, and moreover, if the perception of the degree of certainty is shared between readers. Most studies in this domain assume that certainty can be measured and categorized, though they differ in the number of degrees or categories that are believed to exist, and thus there is no generally-accepted standard for certainty/confidence levels in biomedical text (*Rubinstein et al., 2013*). Wilbur et al. suggested a four category classification: complete certainty/proven fact, high likelihood, low certainty and complete uncertainty. Similarly, *Friedman et al. (1994)* suggest that there are four categories of certainty: high, moderate, low, and no certainty, with an additional "cannot evaluate" category. Aligning with both of these previous studies, *De Waard & Schneider (2012)* encoded four categories of certainty into their Ontology of Reasoning, Certainty, and Attribution (ORCA) ontology as follows: Lack of knowledge, Hypothetical (low certainty), Dubitative (higher, but short of full certainty), Doxastic (complete certainty, accepted knowledge or fact). Other studies have suggested fewer or more certainty categories, and differ in the manner in which these categories are applied to statements.

BioScope (*Vincze et al., 2008*) is a manually-curated corpus, containing 20,924 speculative and negative statements from three sources (clinical free-texts, five articles from FlyBase and four articles from BMC Bioinformatics) and three different types of text (Clinical reports, Full text articles and abstracts). Two independent annotators and a chief linguistic annotator classified text spans as being 'speculative' or 'negative'; other kinds of assertions were disregarded. Thus, the study splits certainty into two categories - speculative, or not.

*Thompson et al. (2011)* apply five meta-knowledge features - manner, source, polarity, certainty, and knowledge type - to the GENIA event corpus (*GENIA Event Extraction , GENIA*). This corpus is composed of Medline abstracts split into individual statements.

Prieto et al. (2020), *PeerJ*, DOI 10.7717/peerj.8871

**Table 1  Comparison of corpora and approaches used in prior investigations into scholarly certainty.**

| | No of annotators | Annotator expertise | Text provenance | Discourse segment source | Approach to automated detection | Number of certainty classification classes | Corpus size | Meta knowledge examined |
|---|---|---|---|---|---|---|---|---|
| *Light, Qiu & Srinivasan (2004)* | 4 | following annotation guidelines | Medline | Abstract | SVM | 3 | 2,093 statements | certainty |
| *Malhotra et al. (2013)* | 3 | following annotation guidelines | Medline | Abstract | Maximum Entropy | 4 | 350 abstracts | certainty |
| *Zerva et al. (2017)* | 7+2 | biomedical | GENIA-MK, BioNLP-ST | Abstract, Text Event | Random Forest classifier + Rule Induction | up to 5 | 652 passages | certainty |
| *De Waard & Maat (2012)* | 2 | publishing | 2 articles | Full text | N/A | 4 | 812 clauses | certainty, basis, source |
| *Friedman et al. (1994)* | 3 | physics | Columbia Presbyterian Medical Database | Free text | Natural Language Processor | 4 | 230 reports | certainty, degree, change, status, quantity, descriptor |
| *Wilbur, Rzhetsky & Shatkay (2006)* | 3+9 | following annotation guidelines | Ten research articles | Full text | N/A | 4 | 101 sentences | focus, polarity, certainty, evidence, and directionality |
| *Vincze et al. (2008)* | 3 | linguistics | Clinical, FlyBase, BMC Bioinfo | Free Text, Full Text, Abstract | N/A | 2 | 20,924 statements | certainty, negation |
| *Thompson et al. (2011)* | 2 | following annotation guidelines | Medline | Abstract | N/A | 3 | 36,858 events | manner, source, polarity, certainty, knowledge type |
| This manuscript | 375 | biomedical | TAC 2014 | Full Text | Neural Network | 3 | 45 statements | certainty |

With respect to certainty annotations, the corpus utilizes a classification system of three certainty levels - certain, probable (some degree of speculation), and doubtful (currently under investigation). Annotation was carried out by two linguistic specialists specifically trained in the meta-knowledge scheme.

*Light, Qiu & Srinivasan (2004)* investigate speculative language in biomedical abstracts. Using Medline abstracts they attempt to distinguish high and low degrees of speculation. Four annotators used rule-matching to classify statements. Using this annotated corpus, they trained a model based on Support Vector Machines (SVM) to generate an automatic classifier. This automatic classifier, therefore, is specifically tasked for speculative statements, and categorizes them in a manner resembling their predefined rule-sets.

*Malhotra et al. (2013)* classify hypotheses (speculative statements) in scholarly text. Three annotators classified speculative statements in Medline abstracts related to Alzheimer's disease using a four-class categorization, with predefined pattern-matching rules for sorting statements into three speculative patterns (strong, moderate, and weak) and a fourth category representing definitive statements. Additionally, they explore several automated methods to distinguish speculative from non-speculative statements.

*Zerva et al. (2017)* use a combination of the BioNLP-ST and GENIA-MK corpora - both of which consist of statements manually-annotated with respect to their certain/uncertain classification (degrees of uncertainty, when available, were merged resulting in a two-category corpus). They applied rule induction combined with a Random Forest Classifier to create an automated binary classification model. This model was run on 260 novel statements, and the output classification was provided to seven annotators who were asked for simple agree/disagree validation of each automated classification. The degree of disagreement between annotators was in some cases surprisingly high, leading the authors to note that "the perception of (un)certainty can vary among users". In a separate experiment, two annotators ranked the certainty of 100 statements on a scale of 1–5. They noted low absolute annotator agreement (only 43% at the statement-level), but high relative agreement (only 8% of statements were separated by more than one point on the 5-point scale). Comparing again to the automated annotations, they found high correlation at the extremes (i.e., scored by the annotators as 1 or 5) but much less correlation for statements rated at an intermediate level, leading them to conclude "...looking into finer-grained quantification of (un)certainty would be a worthwhile goal for future work".

These previous works share important distinctions relevant to the current investigation. First, in every case, the number of certainty categories were predetermined, and in many cases, categorization rules were manually created. Second, in most cases (*Light, Qiu & Srinivasan, 2004*; *Malhotra et al., 2013*; *De Waard & Maat, 2012*; *Wilbur, Rzhetsky & Shatkay, 2006*; *Vincze et al., 2008*; *Thompson et al., 2011*), the work involved a small number of annotators with a knowledge of linguistics, or specifically trained on the annotation system, rather than being experts in the knowledge-domain represented by the statements, but untrained as annotators. Third, in all cases where automated approaches were introduced, the automated task was to distinguish "speculation" from "non-speculation", rather than categorize degrees of certainty. Notably, there was little agreement on the number of categories, nor the labels for these categories, among these

studies. Moreover, the categories themselves were generally not validated against the interpretation of an (untrained) domain-expert reader. As such, it is difficult to know which, if any, of these approaches could be generalized to annotation of certainty within the broader scholarly literature, in a manner that reflects how domain experts interpret these texts.

To achieve this would require several steps: (1) determine if there are clearly delimited categories of certainty that are perceived by readers of scholarly assertions; (2) if so, determine how many such categories exist; and (3) determine the consistency of the transmission of certainty among independent readers (i.e., agreement). If these are determined robustly, it should then be possible to apply machine-learning to the problem of automatically assigning certainty annotations to scholarly statements that would match the perceptions of human readers.

Here, we attempt a data-driven certainty categorization approach. We execute a series of questionnaire-based studies using manually-curated scholarly assertions, in English, to attempt to objectively define categories of perceived certainty. A different set of certainty categories are provided in each questionnaire, and readers are asked to categorize each statement as to their perception of its level of certainty. We use these results to examine the degree of consistency of perceived certainty among readers, and run statistical tests to evaluate the degree to which the categorization system provided in each survey reflects the perception of those asked to use those categories. The categorization system with the highest score—that is, the one that provided the highest level of agreement—was then used to manually create a corpus of certainty-annotated statements. This, in turn, was used to generate a machine-learning model capable of automatically classifying new statements into these categories with high accuracy. We propose that this model could be used within existing text-mining algorithms to capture additional metadata reflecting the nuanced expression of certainty in the original text. Finally, we provide an example of a machine-accessible scholarly publication—a NanoPublication—within which we have embedded this novel contextual certainty metadata.

## MATERIALS & METHODS

### Broad overview

Using TAC Biomedical Summarization Corpus (*Min-Yen, 2018*), we extracted 45 manually-curated scholarly assertions (selection process described below). Using these, a total of 375 researchers in the biomedical domain, in comparable research institutes and organizations, were presented with a series of assertions and asked to categorize the strength of those assertions into four, three, or two certainty categories over the three independently-executed questionnaires. G Index (*Holley & Guilford, 1964*) coefficient analysis was applied to determine the degree of agreement between annotators, as a means to evaluate the power of each categorization system—that is, to test the discriminatory effectiveness of the categories themselves, versus the quality of the annotations or annotators. Prior to performing the statistical analysis, due to the data being compositional in nature, we applied rank transformation to our data. We extracted the essential features of inter-rater

agreement from the questionnaire data using Principal Component Analysis (PCA) to guide our interpretation of the way annotators were responding to the categories presented. The essential number of components identified by PCA were extracted using Horn's parallel analysis, with three categories appearing to be the optimal. We then clustered our collection of statements into these three categories using the k-means algorithm (*Jolliffe, 2011*; *Dunham, 2006*). Finally, we manually generated an author-annotated corpus of statements ("author-annotated", versus a corpus of statements annotated by participants which will be described as "publicly-annotated") using this 3-category system, and applied deep-learning techniques over this corpus to generate an automated classifier model. To evaluate its accuracy, 20-fold Cross-Validation (CV) was used.

## Survey statement selection

The 45 text blocks used in the three surveys were extracted from published articles related to genetic and molecular topics, and were selected from the "Citation text" and "Reference text" portions of the TAC 2014 Biomedical Summarization Track. Each text block contained a sentence or sentence fragment representing a single scholarly assertion that we highlighted and asked the respondents to evaluate, with the remainder of the text being provided for additional context. The 45 assertions were selected using different epistemic modifiers, such as modal verbs, qualifying adverbs and adjectives, references and reporting verbs, which are believed to be grammatical indicators of "value of truth" statements (*De Waard & Maat, 2012*). Given that they are intended to be used for a human survey, with the aim of avoiding annotator fatigue, these were further filtered based on the length of the statement to give preference to shorter ones. These were then separated into groups based on the type of epistemic modifier used, and from these groups, a subset of statements were selected arbitrarily to give coverage of all groups in our final statement corpus (*Prieto, 2019a*; *Prieto, 2019b*). An example survey interface presentation is shown in Fig. 2.

## Survey design

We designed three surveys—S1, S2 and S3—where respondents were asked to assign certainty based on a number of certainty categories—four, two, and three respectively for surveys S1, S2, and S3. All surveys used the same corpus of 45 scholarly assertions. To minimize the bias of prior exposure to the corpus, the surveys were deployed over three comparable but distinct groups of researchers, all of whom will have sufficient biomedical expertise to understand the statements in the corpus.

All participants were presented a series of assertions selected randomly from the 45 in the corpus—15 assertions in S1, increased to 20 assertions in S2 and S3 in order to obtain deeper coverage of the statement set. In S1, participants were asked to assess the certainty of every highlighted sentence fragment based on a 4-point scale with the following response options: High, Medium High, Medium Low, and Low. A 2-point scale was used for S2: Relatively High and Relatively Low and 3-point numerical scale for S3: 1, 2 or 3. In addition to the assessment of certainty, for each assertion, participants were asked to indicate their impression of the basis of the assertion, using a single-answer, multiple-choice question,

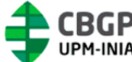 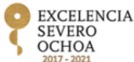 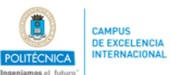 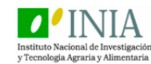

**Scientific Statement:**

Our observations raise the important prediction that many malignancies considered to be non-TK driven because of the absence of a dominant TK mutation may indeed be dependent on TK signaling. It is likely that in different cell types, different PTPs may play roles similar to PTPN12 in suppressing tumorigenesis, possibly by antagonizing different combinations of TKs.

---

**Forget what you know about biology... What do you think is the certainty level expressed by the authors in the statement highlighted in blue?**

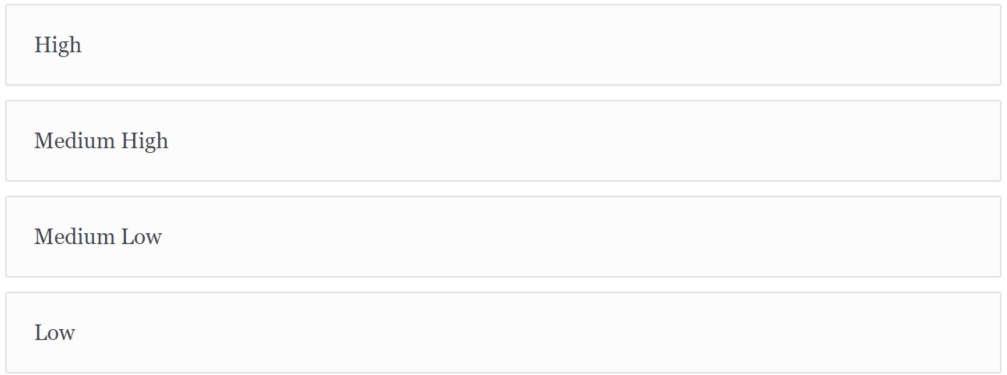

**Figure 2** **Example of the Survey 1 questionnaire interface.** A scholarly assertion is highlighted in blue, in its original context. Participants are asked to characterize the blue assertion, using one of four levels of certainty (High, Medium High, Medium Low or Low).

with the options: Direct Evidence, Indirect Evidence/Reasoning, Speculation, Citation or I don't know.

## Survey distribution and participant selection

Participation in the surveys was primarily achieved through personal contact with department leads/heads of five institutions with a focus on biomedical/biotechnology research. For S1, the majority of participants came from the Centro de Biotecnologia y Genomica de Plantas (UPM-INIA), Spain. It was conducted between November and December of 2016. S2 was executed by members from the Leiden University Medical Center, Netherlands, between November and December of 2017. S3 was conducted between October and November of 2018 by members of the University Medical Center Utrecht, Cell Press and the Agronomical Faculty of Universidad Politécnica de Madrid. Participation was anonymous and no demographic data was collected.

## Survey execution

Participants of the surveys were engaged using the platform Survey Gizmo (S1) or Qualtrics (*Qualtrics, 2017*, S2 and S3)—two online platforms dedicated to Web-based questionnaires.

The change in survey platform was based only on cost and availability; the two platforms have largely comparable interfaces with respect to data-gathering fields such as response-selection buttons and one-question-per-page presentation, with the primary differences between the platforms being aesthetic (color, font, branding).

## Statistical analysis of agreement

We evaluated each survey by quantifying the degree of agreement between participants who were presented the same assertion, with respect to the level of certainty they indicated was expressed by that statement given the categories provided in that survey.

Agreement between participants was assessed by Holley and Guilford's G Index of agreement (*Holley & Guilford, 1964*), which is a variant of Cohen's Weighted Kappa (Kw; (*Cohen, 1968*). Ideally G measures the agreement between participants. It was performed based on the following formula:

$$G = \frac{(\text{Probability Observed}(Po) - \text{Probability by Chance}(Pc))}{1 - Pc}$$

The key difference between Kw and G index is in how chance agreement ($Pc$) is estimated. According to *Xu & Lorber (2014)*, "G appears to have the most balanced profile, leading us to endorse its use as an index of overall interrater agreement in clinical research". For the G index, Pc is defined *a priori*, being homogeneously distributed among categories as the inverse of the number of response categories (*Xu & Lorber, 2014*), thus making Pc = 0.25 for S1; Pc = 0.50 for S2; and Pc = 0.33 for S3.The accepted threshold for measuring agreement and its interpretation has been suggested by *Landis, Richard Landis & Koch (1977)* as follows: 0.00–0.20 = Poor, 0.21–0.40 = Fair, 0.41–0.60 = Moderate, 0.61–0.80 = Substantial, 0.81–1.00 = Almost Perfect. Anything other than the 'Poor' category is considered in other studies to represent an acceptable level of agreement (*Deery et al., 2000*; *Lix et al., 2008*).

## Clustering

As an initial step, due to the compositional structure of the data, it was necessary to perform a transformation of the data prior to Principal Component Analysis (PCA) and correlation statistical analysis (*Mucha, Bartel & Dolata, 2008*). Compositional data are data in which the sum of all components represents the complete set or a constant value (*Mateu-Figueras et al., 2003*). We applied rank transformations (*Baxter, 1995*), due to the presence of essential zeros, indicating an absence of content in a variable (*Foley et al., 2018*). "The presence of zeros prevent us for [sic] applying any measure or technique based upon ratios of components" (*Palarea-Albaladejo, Martín-Fernández & Soto, 2012*). Subsequently, we investigated the ideal number of clusters into which statements group based on the profile of the annotators' responses or inter-survey analyses. To estimate this, Hierarchical Clustering analysis (HCA) and Spearman correlation test were performed to determine certainty category association between questionnaires (Fig. 3), using the shared classified statements in that category as the metric (*Narayanan et al., 2011*; *Campbell et al., 2010*; *Sauvageot et al., 2013*; *Narayanan et al., 2014*); though these constitute conceptually distinct analyses, we represent them in the same chart because the outputs are mutually

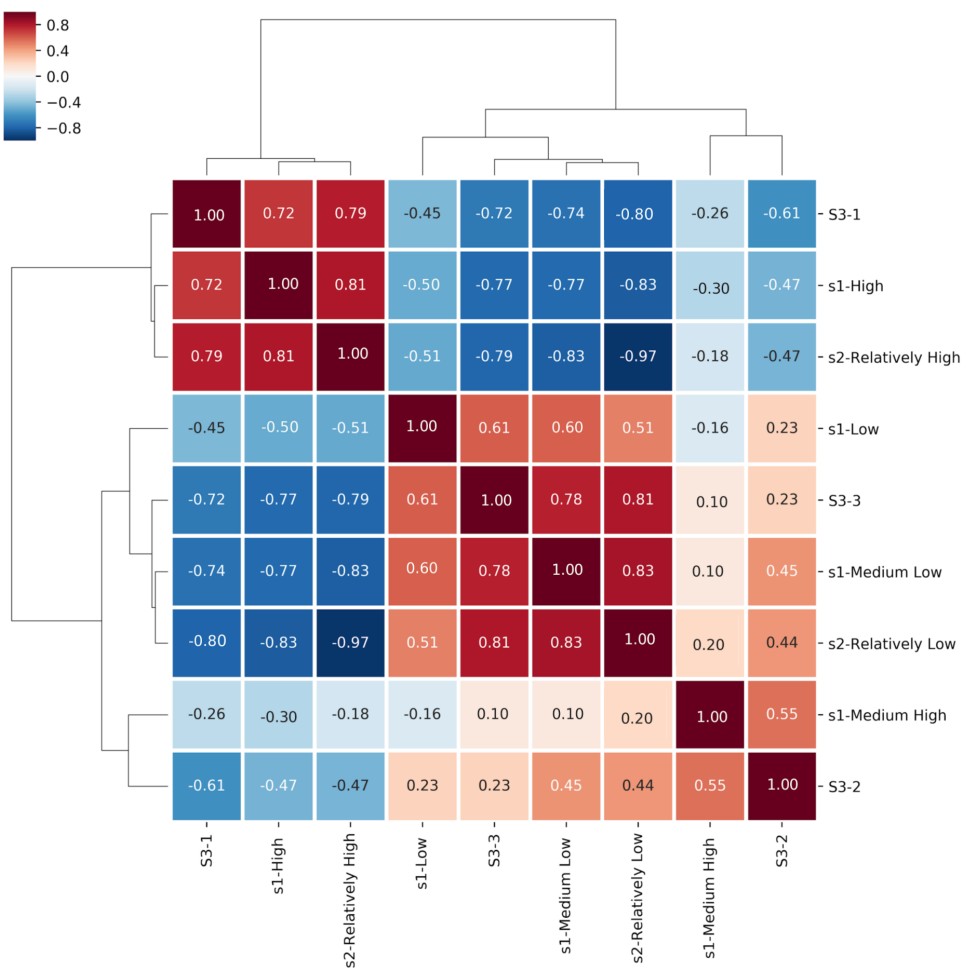

**Figure 3** **Spearman Rank Correlation and hierarchically-clustered heatmap on ranked transformed values comparing the statements assigned to the Certainty Categories among all three questionnaires.** The clustering tree and heatmap are based on participants' responses adjusted by rank transformation from questionnaires S1, S2 and S3. Certainty categories clustered hierarchically. Boxes shows color legend and coefficients based on Spearman's rank-order correlation of the certainty categories.

supportive. Hierarchical clustering analysis (HCA) finds clusters of similar elements, while Spearman correlation coefficient considers the weight and direction of the relationship between two variables. It's worth emphasizing the importance of the rank-based nature of Spearman's correlation. Spearman's formula ranks the variables in order, then measures and records the difference in rank for each statement/variable. Thus, "…if the data are correlated, [the] sum of the square of the difference between ranks will be small" (*Gauthier, 2001*), which should be considered when interpreting the results. Interpretation of Spearman correlation was as follows: Very High >0.9; High ≤ 0.9; Moderate ≤ 0.7; Low ≤ 0.5; and Very Low ≤ 0.2 (*Dunham, 2006*; *Raithel, 2008*). All Spearman correlations are interpreted based on hypothesis testing. To determine the importance of the results, *p*-values were
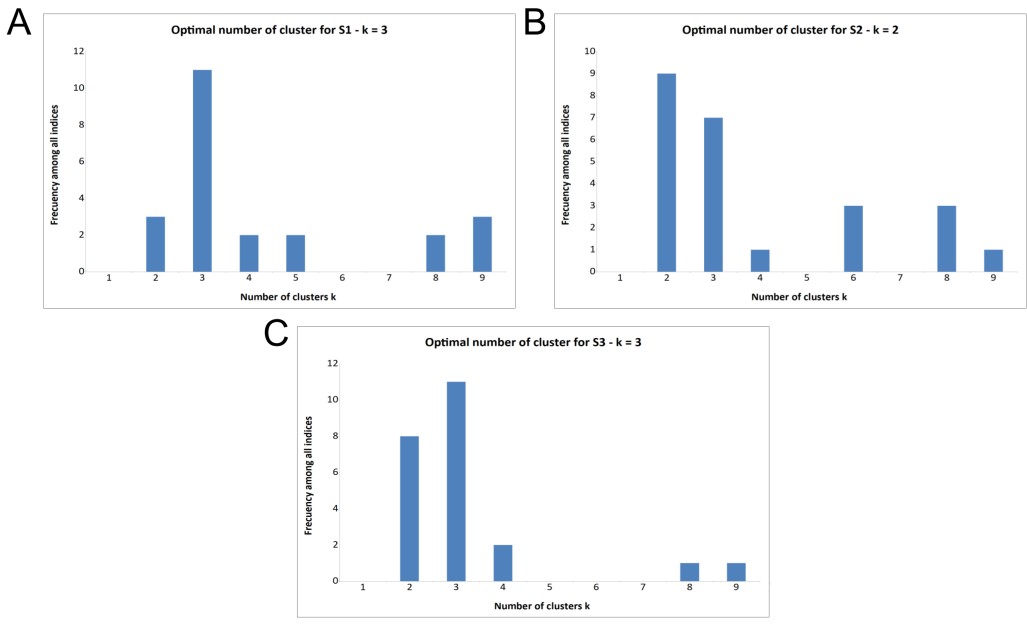

**Figure 4 Majority rule output for deciding optimal number of clusters (*k*) in the three surveys.** (A) Majority rule indicates three clusters for Survey 1. (B) Majority rule indicates two clusters for Survey 2, though there is notable support for three clusters. (C) Majority rule indicates three clusters for Survey 3, with notable support for two clusters.

generated as an indicator of the existence of correlation between certainty categories. Rank, HCA and Spearman values were generated using the python libraries, *seaborn* and *pandas*.

Prior to PCA and cluster analyses, we first adjusted participant's responses using rank transformation from the Python package *pandas*. PCA is a widely used method for attribute extraction to help interpret results. We used PCA to extract the essential features of inter-rater agreement from the questionnaire data (*Campbell et al., 2010*; *Narayanan et al., 2014*). We applied PCA using *scikit-learn* to the result-sets, and utilized K-means from the same python package to identify cluster patterns within the PCA data. These cluster patterns reflect groups of similar "human behaviors" in response to individual questions under all three survey conditions. In the input each statement is represented by the profile of annotations it received from all annotators. The optimal number of components was selected using Horn's parallel analysis, applied to certainty categories on the three different surveys. Detailed output is provided in Figs. S1–S3 of the supplemental information. Our decision to choose three components as the most robust number to capture relevant features of our data is justified in the 'Results'.

To determine the optimal K, (number of statements in each cluster, or cohesion of the clusters), several indices were analyzed using the R package *NbClust* (*Charrad et al., 2014*). *NbClust* provides 30 different indices (e.g., Gap statistic or Silhouette) for determining the optimal number of clusters based on a "majority rule" approach (Fig. 4; *Chouikhi, Charrad & Ghazzali, 2015*). Membership in these clusters was evaluated via Jaccard similarity index comparing, pairwise, all three clusters from each of the three surveys to determine

**Table 2 Jaccard similarity clusters resulting from K-Means applied to questionnaire results.** Jaccard similarity index on k-means results from ranked questionnaire responses. The score is the result from statements labels pairwise comparison. A dash indicates that it is not possible to compare due to differing cluster size.

| | S1-S2 | S1-S3 | S2-S3 |
|---|---|---|---|
| Cluster 1-1 | 0.923 | 0.923 | 0.917 |
| Cluster 1-2 | – | – | – |
| Cluster 1-3 | – | – | – |
| Cluster 2-1 | – | – | – |
| Cluster 2-2 | 0.583 | 0.833 | 0.714 |
| Cluster 2-3 | – | – | – |
| Cluster 3-1 | – | – | – |
| Cluster 3-2 | – | – | – |
| Cluster 3-3 | 0.800 | 0.600 | 0.684 |

which clusters were most alike (Table 2). This provides additional information regarding the behavior of annotators between the three surveys; i.e., the homogeneity of the three identified categories between the three distinct surveys. The *princomp* and *paran* functions in R were utilized to execute PCA and Horn's parallel analysis, respectively. The *PCA* and *KMeans* functions from *scikit-learn* were employed to create the visualizations in Fig. 5 (*Pedregosa et al., 2011*).

## Certainty classification and machine learning model

We addressed the creation of a machine-learning model by considering this task to be similar to a sentiment analysis problem, where algorithms such as Recurrent Neural Network (RNN) with Long Short Term Memory (LSTM) have been applied (*Wang et al., 2016*; *Baziotis, Pelekis & Doulkeridis, 2017*; *Ma, Peng & Cambria, 2018*). A corpus of new statements was extracted from MedScan (Novichkova et al., 2003). An initial filter was applied using the keyword 'that', since this is often indicative of hedging (e.g., "*This result suggests that…*", "*It is thought that…*"). A total of 3,221 statements were manually categorized using the three levels of certainty, based on our familiarity with the classification of the 45 statements in the prior study. A 5-layer neural network architecture was employed to train and validate model performance. Validation was executed using a 20-fold CV scheme, which is considered adequate for a corpus of this size (*Crestan & Pantel, 2010*; *Snow et al., 2008*; *Lewis, 2000*). To design the neural network (NN) model, the Python library *Keras* (*Chollet, 2015*) was utilized, with *TensorFlow* (*Abadi et al., 2016*) as the backend. Precision, recall, F-score and overall accuracy were calculated as additional supporting evidence for classifier performance from a confusion matrix (*Light, Qiu & Srinivasan, 2004*; *Malhotra et al., 2013*; *Zerva et al., 2017*), comprised of the following terms and formulas: True Positive (TP); True Negative (TN); False Positive (FP); False Negative (FN); Precision = TP/(TP+FP); Recall = TP/(TP+FN); F-score = (Precision ×Recall ×2)/( Precision + Recall); Overall accuracy = (TP+TN)/(TP+FP+FN+TN). Finally, we employed Kappa as a commonly-used statistic to compare automated and manual adjudication (*Garg et al., 2019*). Kappa was calculated using the *pycm* python package.
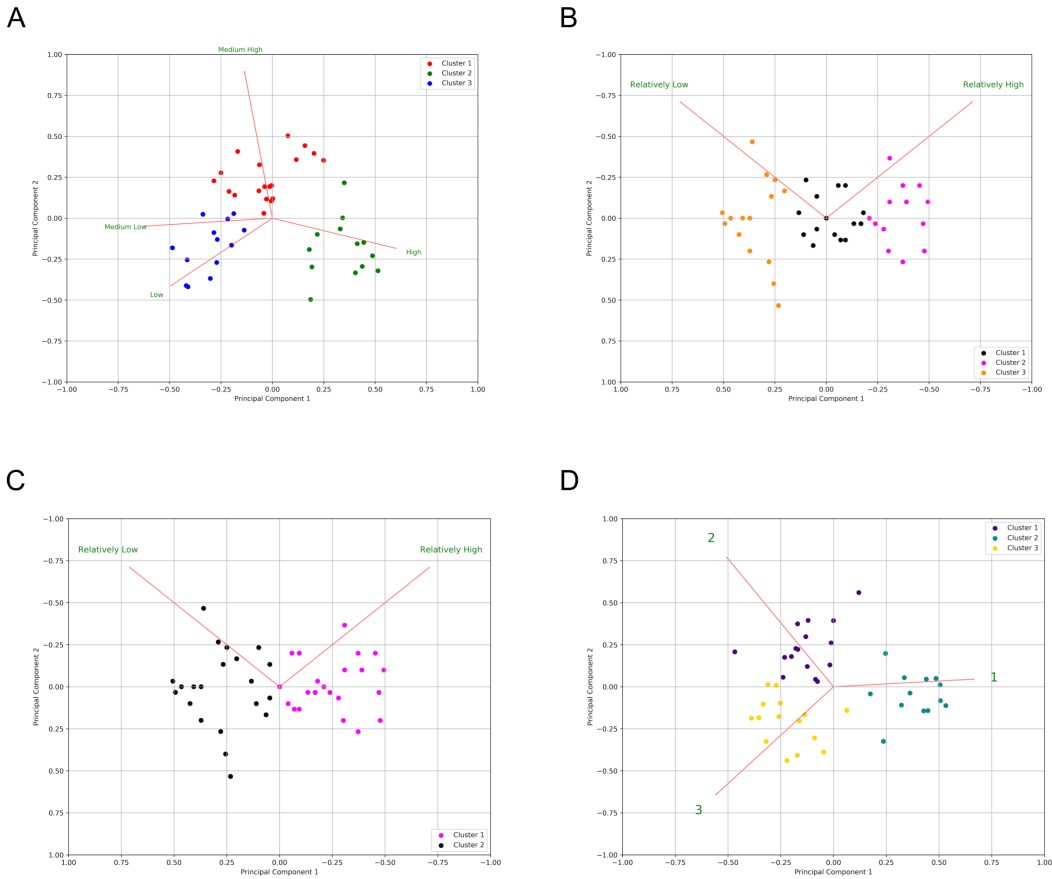

**Figure 5** **Principal component analysis of questionnaire responses in the three surveys.** Bi-plot of certainty level distribution over results from k-means clustering (colors) for: Survey 1 (A), Survey 2 with three clusters (B), Survey 2 with two clusters (C) and Survey 3 (D). Each dot represents a statement. Red lines are the eigenvectors for each component.

All raw data and libraries used are available in the project GitHub, together with Jupyter Notebooks (both R and Python 2.7 kernels) showing the analytical code and workflows used to generate the graphs presented in this manuscript and the supplemental information (*Prieto, 2019a*; *Prieto, 2019b*).

# RESULTS

## Survey participation

Survey 1 (S1) was answered by 101 participants of whom 75 completed the survey (average of 13 responses per participant). Survey 2 (S2) had 215 participants with 150 completing the survey (average of 16 responses per participant). 48 of 57 participants completed the entirety of Survey 3 (S3) (average of 18 responses per participant). All responses provided were used in the analysis. Coverage (the number of times a statement was presented for evaluation) for each of the 45 statements in the corpus was an average of: 29 for S1, 77 for

**Table 3  Categorization consistency of statements (by statement number) for survey S1.**

| Agreement level | High | % of corpus | Medium high | % of corpus | Medium low | % of corpus | Low | % of corpus |
|---|---|---|---|---|---|---|---|---|
| Almost Perfect [0.81–1.00] | 29 | 2.2% | 0 | 0% | 0 | 0% | 0 | 0% |
| Substantial [0.61–0.8] | 25, 27, 30 | 6.6% | 5 | 2.2% | 0 | 0% | 0 | 0% |
| Moderate [0.41–0.6] | 4, 28, 42 | 6.6% | 19, 35, 37, 40, 45 | 11.1% | 21, 36, 44 | 6.6% | 0 | 0% |
| Fair [0.21–0.4] | 3, 15, 22, 38 | 8.9% | 2, 8, 9, 16, 17, 20, 34, 39 | 17.7% | 1, 6, 10, 11, 12, 14, 18, 26, 33 | 20.0% | 0 | 0% |
| Poor [0.2] | 13 | 2.2% | | | | | | |
| Double-Classified | 41, 43 | 4.4% | 7, 23, 24, 31, 32, 41, 43 | 15.5% | 7, 23, 24, 31, 32, | 11.1% | | |

**Table 4  Categorization consistency of statements (by statement number) for survey S2.**

| Agreement level | Relatively high | % of corpus | Relatively low | % of corpus |
|---|---|---|---|---|
| Almost perfect [0.81–1.00] | 25, 27, 28, 29, 30, 41 | 13.3% | 36, 44 | 4.4% |
| Substantial [0.61–0.8] | 3, 15, 22, 38, 40, 42, 43 | 15.5% | 0 | 0% |
| Moderate [0.41–0.6] | 5, 6, 9, | 6.6% | 10, 11, 14, 18, 31, 33, 39 | 15.5% |
| Fair [0.21–0.4] | 4, 37, 45 | 6.6% | 1, 12, 13, 16, 19, 21, 23, 24, 32, 34 | 22.2% |
| Poor [0.2] | 2, 7, 8, 17, 20, 26, 35 | 15.5% | | |
| Double classified | 0 | 0% | | |

S2, and 23 for S3. The summary of the k-means clustering and Jaccard Similarity results over all three surveys are shown in Fig. 3 and Table 2.

### Survey 1

In S1, all statements except statement #13, scored at or above the minimum agreement ($G = 0.21$; "Fair" degree of agreement on the (*Landis, Richard Landis & Koch, 1977*) scoring system). Seven of 45 statements (16%) showed inter-annotator agreement achieving statistically-significant scores in two certainty categories simultaneously. Table 3 shows the distribution of statements among certainty categories and agreement levels. 11 of 45 statements (24%) were classified as High certainty; 14 of 45 statements (31%) were Medium High; Medium Low was represented by 12 of 45 statements (27%); and the Low certainty category did not produce inter-annotator agreement for any statement

### Survey 2

Disposition of Certainty categories and agreement levels for S2 are shown in Table 4. Seven of the 45 statements (16%) did not achieve significant agreement for any certainty level. Relatively High was selected for 19 of 45 statements (42%). The remaining statements (19/45; 42%) were selected as Relatively Low.

### Survey 3

Table 5 summarizes the levels of agreement and certainty classifications observed in S3. Categories were ranked numerically from 1 (the highest level of certainty) to 3 (the lowest

**Table 5** Categorization consistency of statements (by statement number) for survey S3.

| Agreement level | Category 1 | % of corpus | Category 2 | % of corpus | Category 3 | % of corpus |
|---|---|---|---|---|---|---|
| Almost Perfect [0.81–1.00] | 3, 15 | 4.4% | 0 | 0% | 0 | 0% |
| Substantial [0.61–0.8] | 27, 28, 29, 38, 42 | 11.1% | 0 | 0% | 0 | 0% |
| Moderate [0.41–0.6] | 4, 25, 30, 41, 43 | 11.1% | 2, 16, 17, 23, 26, 33, 34, 35, 37, 40 | 22.2% | 0 | 0% |
| Fair [0.21–0.4] | 22 | 2.2% | 1, 6, 8, 9, 10, 11, 12, 13, 18, 19, 20, 31, 32, 45 | 31.1% | 21, 24, 36, 44 | 8.8% |
| Poor [0.2] | 5, 7, 14, 39 | 8.8% | | | | |
| Double classified | 0 | 0% | | | | |

level of certainty). Minimum agreement ($G = 0.21$) or superior was observed in 41 of 45 statements (91%) with no doubly-classified statements, indicating little evidence of annotator-perceived overlap between the presented categories. Four of 45 statements (9%) did not obtain agreement for any certainty category. Category 1 was selected for 13 of 45 statements (29%). 24 of the total of 45 (53%) were chosen with level of Category 2. Finally, Category 3 was selected for four out of 45 statements (9%).

## Clustering

As shown in Fig. 3, HCA and Spearman correlation revealed three primary clusters when executed over the three surveys combined. Considering only the horizontal axis, the leftmost cluster includes S3-Category1, S1-High and S2-Relatively High. Numbers inside the squares of this cluster show significant Spearman correlation (**S1-High/S2-Relatively High:** $r = 0.81$, *p*-value $< 0.001$; **S1-High/S3-1:** $r = 0.72$, *p*-value $< 0.001$; **S2-Relatively High/S3-1:** $r = 0.79$, *p*-value $< 0.001$). The second branch of the HCA is split into two main sub-trees, including the center and right regions of Fig. 3. The cluster in the center side of the figure, differentiated by excellent Spearman correlation, contains S1-Medium Low, S2-Relatively Low and S3-Category3 (**S1-Medium Low/S3-3:** $r = 0.78$, *p*-value $< 0.001$; **S2-Relatively Low/S3-3:** $r = 0.81$, *p*-value $< 0.001$; **S1-Medium Low/S2-Relatively Low:** $r = 0.83$, *p*-value $< 0.001$). Finally, the smaller cluster identified by HCA on the right side of Fig. 3 comprises S1-Medium High and S3-Category2 with moderate Spearman correlation ($r = 0.55$, *p*-value $< 0.001$), confirming that a third certainty category has sufficiently strong support.

Supporting the previous cluster tests, using the majority rule approach, based on the indices that were available in *NbClust*, the results (Fig. 4) indicate that:

- 11 indices proposed 3 as the optimal number of clusters for the results of S1 (Fig. 4A)
- 9 indices proposed 2 as the optimal number of clusters for the results of S2 (Fig. 4B)
- 7 indices proposed 3 as the optimal number of clusters for the results of S2 (Fig. 4B)
- 11 indices proposed 3 as the optimal number of clusters for the results of S3 (Fig. 4C).

Note that, surprisingly, the second-most optimal number of clusters for Survey 2 was three (Fig. 4B), despite S2 having only two possible responses. This will be discussed further in the 'Discussion' section.

**Table 6  Analysis of Principal Components of survey S1.**

| Principal Components: | Comp.1 | Comp.2 | Comp.3 | Comp.4 |
|---|---|---|---|---|
| High | 0.597 | 0.200 | 0.336 | 0.700 |
| Medium High | −0.126 | −0.885 | 0.418 | 0.161 |
| Medium Low | −0.610 | 0.033 | −0.382 | 0.694 |
| Low | −0.505 | 0.418 | 0.753 | −0.050 |
| Component variances | 2.276 | 1.137 | 0.386 | 0.200 |
| Proportion of Variance | 0.569 | 0.284 | 0.097 | 0.050 |
| Cumulative Proportion | 0.569 | 0.853 | 0.950 | 1.000 |

Component Variances (row 6), Proportion of Variance (row 7) and Cumulative Proportion (row 8) are summarized in Table 6 for S1, for each principal component. Table 6 additionally supplies the information to explain each component and its relative weighting, requisite to understanding all components. Horn's parallel analysis on S1 and S3 retained optimally two factors, though three factors was also within acceptable boundaries. S2 also retained two factors. Given the results of the cluster analysis (Fig. 4), and given the more robust separation of, and cohesion within categories in the third survey, we believe that the optimal number of components to retain is three. Detailed output is provided in Fig. S1–S3. The first three components explain 95% of the variance of the data. Figure 5A shows the graph resulting from a principal component analysis (PCA) of responses to statements from S1, clustered by K-Means (colored dots). Red lines represent the eigenvectors of each variable (here the certainty categories) for PC1 against PC2. A coefficient close to 1 or -1 indicates that variable strongly influences that component. Thus, the High category has a strong influence on PC1 (0.59), Medium High negatively influences PC2 (−0.88), and Medium Low and Low have a notably strong negative relationship with PC1 (−0.61 and −0.50, respectively). Additionally, Low strongly influences PC3 in a positive manner (0.75). The same approach was followed for S2 and S3, with the results shown in Fig. 5B, C, and D. For survey S2, we show the K-Means clustering results for both a three-cluster solution (Fig. 5B), and a two-cluster solution (Fig. 5C).

## Machine learning

The corpus of 3,221 author-annotated statements was used to train a 5-layer NN model. This was validated using 20-fold CV due to the size of the dataset, with the result indicating that it achieved 89.26% accuracy with a standard deviation of 2.14% between folds. A test of its performance relative to the highest-scoring dataset (Survey 3, majority rule classification of the publicly-annotated 45 statements) showed 82.2% accuracy (see Table 7, right). A further test was done to validate the author-categorized corpus compared to the publicly annotated dataset (see Table 7, left). Majority rule vs. the author's classification gave a kappa value of 0.649 (substantial), while comparison with the model's classification gave a kappa of 0.512 (moderate).
**Table 7  Performance of the neural network model on the 45 publicly-annotated statements.**

| | S3 Majority Rule vs. Author's Classification | | | | S3 Majority Rule vs. Model's Classification | | | |
|---|---|---|---|---|---|---|---|---|
| | Precision | Recall | F-Score | Overall accuracy | Precision | Recall | F-Score | Overall accuracy |
| Category 1 | 0.857 | 0.923 | 0.889 | 0.933 | 0.786 | 0.786 | 0,786 | 0.867 |
| Category 2 | 0.692 | 0.947 | 0.800 | 0.800 | 0.778 | 0.808 | 0,792 | 0.756 |
| Category 3 | 1.000 | 0.385 | 0,555 | 0.822 | 0.250 | 0.200 | 0,222 | 0.844 |
| Average | 0.849 | 0.751 | 0.748 | 0.851 | 0.604 | 0.598 | 0,600 | 0.822 |
| Confusion Matrix | | | | | | | | |

| | 1 | 2 | 3 | | | 1 | 2 | 3 |
|---|---|---|---|---|---|---|---|---|
| 1 | 12 | 1 | 1 | | 1 | 11 | 3 | 0 |
| 2 | 1 | 18 | 7 | | 2 | 2 | 21 | 3 |
| 3 | 0 | 0 | 5 | | 3 | 1 | 3 | 1 |

| Kappa | 0.649 | 0.512 |
|---|---|---|

# DISCUSSION

## Evidence to support three levels of certainty in scholarly statements

In S1, we began with a four-category classification system, since this is the highest number presumed in earlier studies (Zerva et al. used a 5-point numerical scale, but we do not believe they were proposing this as a categorization system). In the absence of any agreed-upon set of labels between these prior studies, and for the purposes of asking untrained annotators to categorize scholarly statements, we labelled these categories High, Medium High, Medium Low and Low. The results of this survey revealed statistically significant categorization agreement for 37 of the 45 statements (82% of total), with seven statements being doubly-classified and one statement showing poor inter-annotator agreement, for a total of eight 'ambiguous' classifications. The G index (*Holley & Guilford, 1964*) with only four categories is small, and the statistical probability of chance-agreement in the case of ambiguity is therefore high, which may account for the high proportion of doubly-classified statements. Interestingly, the category Low was almost never selected by the readers. We will discuss that observation in isolation later in this discussion; nevertheless, for the remainder of this discussion we will assume that the Low category does not exist in our corpus of ~3,200 author-annotated statements, and will justify that in later detailed arguments.

With respect to the categories themselves, the category of High had robust support using the G index statistic, indicating that it represents a valid category of certainty based on agreement between the annotators on the use of that labelled category. Support for the other two, medium-level, categories was less robust. This could be interpreted in two ways - one possibility is that these two categories are not distinct from one another, and that readers are selecting one or the other "arbitrarily". This would suggest that there are only two certainty categories used in scholarly writing. The other option is that the labels assigned to these two non-high categories do not accurately reflect the perception of the reader, and thus that the categorizations themselves are flawed, leading to annotator confusion.

In Survey 2, with only two categories (Relatively High and Relatively Low), statistical support for these two categories was evident, but deeper examination of the results suggests

that these categories may still not accurately reflect the reader's perception. For example, seven of the 45 statements (16%) showed no inter-annotator agreement. Of the remainder, Table 4 shows a clear pattern of association between the strength of certainty perceived by the reader, and the degree to which the readers agreed with one another. Effectively, there was greater agreement on the categorization of high-certainty statements than low-certainty statements. This mirrors the observations from Survey 1, where the category High generated the highest levels of agreement among annotators. Since this binary categorization system lacks an intermediate category, the Pc index in this survey is 0.5, meaning that agreement by chance is high. It appears that statements that would have been categorized into a middle class from Survey 1 became distributed between the two Survey 2 categories, rather than being categorized uniformly into the lower category. This would indicate that the two-category explanation for Survey 1 is not well-supported, and possibly, that the labelling of the categories themselves in both Survey 1 and Survey 2 confounds the analysis and does not reflect the perception of the reader. In other words, the category High/Relatively High seems to match a perception that exists in the minds of the readers, but the categories Medium High (S1), Medium Low (S1) and Relatively Low (S2) might not correspond to the perception of the readers for the lower certainty statements, which is why they are less consistent in the selection of these categories.

To reveal patterns of annotator behavior within and between surveys we utilized a variety of clustering approaches (Figs. 3 and 4). That there are three, rather than two or four, categories is supported by the hierarchical clustering of all three surveys and a majority rule approach, shown in Fig. 3 (see clusters along the top edge) and Fig. 4. Figure 3 reveals three primary clusters in the data, where high is strongly differentiated from non-high categories. The output from NbClust's "majority rule" approach to selecting the optimal number of clusters based on the number of statements was executed on individual surveys. The results for S1 and S2 are shown in Fig. 4A and Fig. 4B. The majority rule indicates that there were three discernable clusters in S1. Survey 2 was assessed by the 30 NbClust indices (*Charrad et al., 2014*) (Fig. 4B). Surprisingly, we found that, while nine indices recommended only two clusters, seven indices suggested that there were three clusters. Since a cluster represents a pattern of categorization-behavior among all evaluators, we take these results as further indication that there are three discernable annotator responses when faced with a certainty categorization task.

To further explore the meaning of these clusters, we executed a feature reduction analysis using Principal Components. The PCA of Survey 1 revealed three primary components accounting for ∼95% of the variability. The main component, accounting for more than half (∼57%) of the variation, is characterized by a strong positive influence from the category labelled High, and a negative influence from the categories labelled Medium Low and Low. This lends support to our earlier interpretation that there is little ambiguity among annotators about what statements are classified as highly certain, and moreover, when faced with a high-certainty statement annotators will almost never select one of the low categories. The second and third components (accounting for ∼28% and ∼10% respectively) are more difficult to interpret. Component 2 is characterized by a strong negative influence from the category Medium High; Component 3's "signature" is distinguished by a positive

influence from the category Low, though as stated earlier, this category was rarely selected and showed no significant agreement among annotators, making this difficult to interpret. The lack of clarity regarding the interpretation of these second and third components may reflect ambiguity arising from the labelling of the non-high certainty categories in the questionnaire; effectively, the words used for the labels may be confusing the readers, and/or not aligning with their impressions of the statements.

In an attempt to gain additional evidence for a three-category classification system, we undertook a third survey (S3) in which the reader was offered three categories, ordered from higher to lower, but with numerical labels (1, 2, or 3). The rationale for this was twofold. First, we could not think of three suitable labels that would not inherently bias the results (for example, 'high', 'medium', and 'low' would not be suitable because we have already determined that the category 'low' is almost never selected). In addition, we wished to know if category labels were a potential source of bias, and therefore more semantically neutral labels might lead to a stronger correspondence between the annotators. Indeed, Survey 3 generated the most consistent agreement of the 3 questionnaires, where only four of the 45 statements did not meet the cutoff level for annotator agreement, and none were doubly-classified. It is not possible to disambiguate if this enhanced agreement is due to the annotators being presented with a ''correct'' number of categories, or if it supports the suggestion that the presentation of meaningful (but non-representative) category labels caused annotators to behave inconsistently in S1 and S2, or perhaps a combination of both. As with S1, NbClust's ''majority rule'' proposes three clusters for S3 (Fig. 4C).

In Fig. 3 we present the correlation matrix to show how the categories relate to one another between the three surveys, using Spearman Correlation. High (S1) is clearly correlated with Relatively High (S2) and Category 1 (S3). Medium Low (S1), Relatively Low (S2) and Category 3 (S3), are also highly correlated. Low (S1) only has moderate correlation with Medium Low (S1), Relatively Low (S2) and Category 3 (S3). The intermediate values Medium High (S1) and Category 2 (S3) are found on the negative side of Principal Component 1 (Figs. 5A & 5D), which supports the interpretation that a High certainty category is strongly supported, and strongly distinct from other categories. The non-high categories appear as distinct blocks within the correlation matrix, but with more ambiguity or inconsistency, though the Jaccard similarity index was sufficient to support the existence of these two lower-certainty categories. Additionally, the clusters identified by the Spearman analysis (3 clusters) are supported by the results of the HCA analysis (3 branches).

One general source of inconsistency we noted in the data could be described as a ''tendency towards the middle''. When a category is removed, statements from that category tend to distribute to adjacent categories. We presume this reflects some form of ''central tendency bias'', a behavioral phenomenon earmarked as a preference for selecting a middle option. (Hollingworth, 1910; Huttenlocher, Hedges & Vevea, 2000; Duffy et al., 2010). Nevertheless, this did not appear to be sufficiently strong in this investigation to mask the detection of distinct clusters of categorization behavior.

In summary, the results suggest that there are three categories of certainty in the minds of the readers of scholarly assertions. One category is clearly distinguished as representing high-certainty statements. The other two categories, representing non-high certainty

statements, are also well distinct from one another in the minds of the annotators, however, seem to not be reflected well by the labels "moderately/relatively + high/low". Nevertheless, they do appear to represent a higher-to-lower spectrum, since the replacement of textual labels with a numerical range resulted in stronger annotator agreement about these two lower categories.

## The absence of a Low certainty category

Several studies that preceded this one (*Friedman et al., 1994*; *Wilbur, Rzhetsky & Shatkay, 2006*; *De Waard & Schneider, 2012*) suggested four categories of certainty, with one of those being a category that would represent the lowest certainty. In this study, we identify only three. The category that seems to be absent from our data is this lowest category - generally described as "no knowledge" in these three precedent studies. We examined our corpus and, given the grammatical cues suggested by *De Waard & Maat (2012)* we identified two statements in our corpus that, by those metrics, should have scored in the Low category. Those are Statement 3, "*However, this was not sufficient for full blown transformation of primary human cells, which also required the collaborative inhibition of pRb, together with the expression of hTERT, RASV12.*", and Statement 4, "*Hence, the extent to which miRNAs were capable of specifically regulating metastasis has remained unresolved.*" Looking at the results in Tables 3–5, these two statements were annotated with considerable agreement as high-certainty statements - the opposite of what would have been predicted. One explanation for this is that the statements are making a negative claim, with high certainty, and thus are being categorized as high-certainty assertions by our annotators. If that is the case, then the category of "no knowledge" may not be a category that lies anywhere on the spectrum of certainty, and may reflect a distinct feature of scholarly communication discourse, or (more likely) a combination of the meta-knowledge facets of certainty and polarity.

## Application of this categorization system

As indicated in the Introduction, a primary motivation for this study is its application to the automated capture of metadata related to the certainty being expressed in text-mined scholarly assertions, or to identify or monitor 'hedging erosion'. To demonstrate how the outcomes of this study can be applied, we have used the data described here to generate, by machine-learning, an automated certainty classifier capable of assigning new scholarly statements into one of the three certainty categories. Two exemplar outputs from this classification system are shown in Figs. 6 and 7. Figure 6 shows three sets of statements, color-coded by the category of certainty detected by our classifier - green (Category A, associated with High certainty), orange (Category B, non-high/moderate), and red, (Category C non-high/low). Two citation chains relate to the accumulation of beta-APP in muscle fibers of Alzheimer's Disease patients (Figs. 6A & 6B), while Fig. 6C shows a longer citation chain identified by Greenberg as being problematic with respect to 'citation-distortion' (*Greenberg, 2009*). The panels reveal that the degree of certainty can change through citation, becoming higher (Figs. 6A & 6B). Figure 6C reveals a similar trend toward increasing certainty, with the exception of one author who used a clearly

**A**

*"We have previously demonstrated that accumulation of AβPP epitopes precedes other abnormalities in IBM muscle fibers" (Askanas et al., 2000b)*

*"βAPP accumulation is considered to play a major role in the pathogenesis of IBM and AD and is thought to precede other changes in both diseases"(Askanas et al., 1996)*

*"Those muscle fibers, widely prevalent in our one case of hereditary IBM, may represent early changes of IBM and therefore be analogous to the finding in AD brains where PAP accumulations in the "diffuse" Congored-negative plaques seem to represent early changes"(Askanas, Engel & Alvarez, 1992)*

**B**

*"We have previously demonstrated that accumulation of AβPP epitopes precedes other abnormalities in IBM muscle fibers" (Askanas et al., 2000b)*

*"Increased βAPP-mRNA and increased accumulation of βAPP epitopes appear to precede other abnormalities in IBM muscle fiber" (Askanas et al., 1997a)*

*"One possibility is that one protein is accumulated first, due to excessive synthesis, e.g., excessive transcription of mRNA in the IBMs is known for beta APP"(Askanas & Engel, 1995)*

**C**

*Recently it was reported that s-IBM vacuolated muscle fibers, and those in some other vacuolar myopathies, contain a marker of autophagosomes, but only in s-IBM is it colocalized with AβPP[18]. (Askanas and Engel 2007)*

*Overexpression of amyloid precursor protein (APP) and subsequent accumulation of cleaved fragments including β-amyloid in vacuolated muscle fibers is considered a central mechanism in the pathogenesis of s-IBM.[2] (Lünemann et al. 2007)*

*it is now established that Aβ/AβPP is also abnormally accumulated in muscle fibers of s-IBM patients, where they are considered to play an important pathogenetic role[4,5,6,7] (Askanas and Engel 2006)*

*A possibility that excessive accumulation of AβPP/Aβ induces inflammation has been proposed by us and by others.[1-3,7,10] (Askanas and Engel 2003)*

*Deposition of the Aβ fragment of the amyloid precursor protein is a feature of affected muscle in IBM (see below) and it has been shown that muscle cells can secrete Aβ.[10] Interaction of Aβ with muscle cells in turn can stimulate IL-6 production by these cells [19]… (Mastaglia et al. 2003)*

*However, in some abnormal muscle fibers in IBM, the accumulation of βAPP appears to extend outside the muscle fiber boundary. This may have been attributable to a fragility of the fiber's surface membrane, which could have been transiently broken.[25] (Baron et al. 2001)*

**Figure 6** **Automated classification of scholarly assertions related to the accumulation of beta-APP protein in muscle fibres, color coded as green (Category A: highest certainty), orange (Category B: medium certainty) and red (Category C: lowest certainty).** (A and B) Two citation chains showing that the degree of certainty expressed in the most recent statement is higher than that in the cited text. (C) A selection of statements identified by *Greenberg (2009)*, as being potentially indicative of 'citation distortion'. In this panel, there is a general trend to higher certainty over time, with the exception of an early high-certainty statement by Mastaglia in 2003 (second row from the bottom).

high-certainty assertion four years before others in the community expressed the same idea with certainty.

Figure 7 demonstrates how this certainty classification could be used to enhance the quality of machine-extracted information. The figure shows a block of machine-readable information following the NanoPublication model for scholarly publishing.

```
@prefix this: <http://w3id.org/nanopub_mario/CertID_3ab55c9c-2321-11ea-b65c-fc4dd447acf2> .
@prefix sub: <http://w3id.org/nanopub_mario/CertID_3ab55c9c-2321-11ea-b65c-fc4dd447acf2#> .
@prefix void: <http://rdfs.org/ns/void#> .
@prefix dcterms: <http://purl.org/dc/terms/> .
@prefix dcelem: <http://purl.org/dc/elements/1.1/> .
@prefix np: <http://www.nanopub.org/nschema#> .
@prefix pav: <http://swan.mindinformatics.org/ontologies/1.2/pav/> .
@prefix prov: <http://www.w3.org/ns/prov#> .
@prefix xsd: <http://www.w3.org/2001/XMLSchema#> .
@prefix rdfs: <http://www.w3.org/2000/01/rdf-schema#> .
@prefix rdf: <http://www.w3.org/1999/02/22-rdf-syntax-ns#> .
@prefix dcat: <http://www.w3.org/ns/dcat#> .
@prefix schema: <https://schema.org/> .
@prefix thispub: <https://dx.doi.org/10.1371/journal.pone.0073940#> .
@prefix orca-x: <http://w3id.org/orca-x#> .

sub:Head {
    this: np:hasAssertion sub:assertion ;
    np:hasProvenance sub:provenance ;
    np:hasPublicationInfo sub:pubinfo ;
    a np:Nanopublication .
}

sub:assertion {
    orca-x:asserts-3ab55c9c-2321-11ea-b65c-fc4dd447acf2 rdf:singletonPropertyOf orca-x:asserts .
    thispub: orca-x:asserts-3ab55c9c-2321-11ea-b65c-fc4dd447acf2 "Consequently miRNAs have
been demonstrated to act either as oncogenes (e.g., miR-155, miR-175p and miR-21) [15,16] or
tumor suppressors (e.g., miR-34, miR-15a, miR-161 and let-7)" .
    orca-x:asserts-3ab55c9c-2321-11ea-b65c-fc4dd447acf2 orca-x:hasConfidenceLevel orca-
x:CategoryA .
}

sub:provenance {
    sub:assertion dcterms:author "Certainty Classifier" ;
    dcterms:title "Automated Certainty Classification of Statement from
https:dx.doi.org/10.1371/journal.pone.0073940" ;
    dcterms:license <https://creativecommons.org/publicdomain/zero/1.0/> ;
    schema:identifier this: ;
    dcat:distribution sub:assertion ;
    prov:wasDerivedFrom sub:_1 .

    sub:_1  dcelem:format "application/pdf" ;
    a void:Dataset , dcat:Distribution ;
    dcat:downloadURL <https://dx.doi.org/10.1371/journal.pone.0073940> .

}

sub:pubinfo {
    this: dcterms:created '2019-12-20'^^xsd:date ;
    dcterms:rights <https://creativecommons.org/publicdomain/zero/1.0> ;
    dcterms:rightsHolder <https://orcid.org/0000-0002-9416-6743> ;
    pav:authoredBy "Mario Prieto" , <https://orcid.org/0000-0002-9416-6743> ;
    pav:versionNumber "1" ;
    prov:wasGeneratedBy "Mario Prieto's Certainty Classifier" .
}
```

**Figure 7**  **An exemplar prototype NanoPublication including certainty annotations.** The figure shows how certainty classifications could be used as additional, and important metadata when added to text-mining pipelines. A NanoPublication is a machine-readable representation of a scholarly assertion, carrying with it all of its provenance. In this exemplar (hypothetical) NanoPublication for statement #29 in this study, the concept being asserted (that microRNA mir-155 has 

**Figure 7 (…continued)**
the function of a Tumor Suppressor) is captured using ontologically-based concepts in the "assertion" block of the NanoPublication (red text), together with the proposed annotation of that statement's certainty category (blue text) being ORCA-X Category A. This could be used, for example, to filter assertions based on the degree of certainty they express. The final block, pubinfo, contains authorship, license, and citation information for the NanoPublication itself, expressing the terms of usage of this metadata, and who to cite (green text). This entire structure can be interpreted by automated agents, and fully complies with the FAIR Data Principles.

The sentence which has been extracted in this exemplar is from the article with DOI '10.1371/journal.pone.0073940', and the specific sentence "Consequently miRNAs have been demonstrated to act either as oncogenes (e.g., miR-155,miR-17 −5p and miR-21) or tumor suppressors (e.g., miR-34,miR-15a,miR-16 −1 and let-7)". Following the rules of NanoPublications, a single scholarly assertion is captured - in this case, that "miR-34 has the function of tumor suppressor" (red text). The provenance block contains information showing the degree of certainty being expressed (Category A, which maps to the highest certainty category in our classifier; blue text). Finally, there is a block of citation information regarding the NanoPublication itself, such that the author of the certainty classification can be properly cited (green text).

## Tools for researchers, authors, reviewers, and data miners

As discussed in the introduction, researchers may lack the knowledge required to assess the legitimacy of claims that are not directly in their domain, or may be unaware of the history of a claim if they have not followed a citation chain to its roots. Similarly, when acting as peer reviewers, there is little tooling to assist them in evaluating the validity of assertions in the submitted manuscript or funding proposal. In parallel with research into automated identification of reference-spans (*Saggion, AbuRa'ed & Ronzano, 2016*), the availability of a certainty classifier would make it possible to automate the creation of annotated citation chains such as shown in Fig. 6. Reviewers could then use these to determine if a claim was being made with unusually high (or low) certainty—like the Magstalia statement from 2003, shown in Fig. 6C—and thus enhance the confidence of their reviews. Similarly, such tools could become an important part of the scholarly planning process. During the preparation of a paper or proposal, researchers could be made aware of dubious assertions, and avoid relying on these as the bases for their hypothesis. In the context of automated data mining, assuming that incremental steps towards certainty should be associated with the existence of supporting data, the automated detection of "certainty inflection points" could be used by data mining algorithms to identify the specific dataset containing data supporting (or refuting) a given claim. Together with the use of certainty classification in the context of text-mining discussed above, the use of such a classification system may become an important part of the scholarly publishing lifecycle.

## Future investigations to elucidate perceptions of certainty

A variety of future studies could provide additional insight into how researchers communicate and perceive certainty. The results presented here seem to suggest that words like "medium" and "low" do not align well with the perception held by researchers

as they read statements that fall into non-high certainty categories. Future studies could extract additional information in the questionnaire, such as questions related to the basis upon which an assertion was made (e.g., speculation, direct or indirect observation, etc.), as it may be that the distinction between the non-high certainty categories is being made based on other kinds of implicit information, rather than being specifically "medium" or "low" expressions of certainty. It would also be interesting to capture demographic information, to determine if perception of certainty changes as a researcher becomes more experienced, if it differs between different linguistic groups, or if it is associated with other demographic variables

## CONCLUSIONS

This study attempted to derive a data-driven certainty classification system, using statements from scholarly literature in the biological sciences. We found support for three categories of certainty within the dataset of 45 scholarly statements we selected. These consisted of one well-defined High Certainty category, and two non-high certainty categories that were seemingly not well-described using textual labels, but were clearly distinguishable from one another using statistical algorithms. We suggest that a fourth category described in previous studies—best described as "lack of information"—likely does not belong in the same categorization system, and is likely a measure of a different discourse feature than "certainty". Finally, we show how this categorization system could be used to capture key contextual information within text-mining pipelines, to improve the quality of automated information capture. Work on the machine-learning models leading to such an automated classifier are well underway, and we demonstrate that they are already showing a high degree of accuracy, indicating that machines may be capable of detecting and distinguishing the subtle linguistic cues of certainty that we have observed in this study. While this study was limited to biomedical statements, and thus may be applicable only in this domain, it nevertheless seems likely that the results will be more generalizable, at least within the sciences where these kinds of grammatical structures are commonly used.

## ACKNOWLEDGEMENTS

We wish to thank all of the anonymous volunteers who donated their time to answering these questions to the best of their ability. We wish to thank Leiden University for hosting us during a student exchange, and for providing free access to their Qualtrics questionnaire platform. The authors would like to acknowledge Dr. Ron Daniel Jr. for useful discussions around survey design and the staff at Cell Press for allowing use of their Boston offices and providing feedback on the surveys. We would also like to thank the Foundation DTL DP (Data Projects) for their support of this initiative.

### Funding

This work has been funded by the Isaac Peral/Marie Curie cofund with the Universidad Politécnica de Madrid, and the Spanish Ministerio de Economía y Competitividad grant number TIN2014-55993-RM and the ''Severo Ochoa Program for Centres of Excellence in R&D'' from the Agencia Estatal de Investigación of Spain (grant SEV-2016-0672 (2017-2021) to the CBGP). Additional support was provided by the Consejo Social de la Universidad Politécnica de Madrid. The funders had no role in study design, data collection and analysis, decision to publish, or preparation of the manuscript.

### Grant Disclosures

The following grant information was disclosed by the authors:
Isaac Peral/Marie Curie cofund with the Universidad Politécnica de Madrid.
Spanish Ministerio de Economía y Competitividad: TIN2014-55993-RM.
Severo Ochoa Program for Centres of Excellence in R&D.
Agencia Estatal de Investigación of Spain: SEV-2016-0672.
Consejo Social de la Universidad Politécnica de Madrid.

### Competing Interests

Helena Deus is employed by Elsevier Inc. Anita de Waard is Vice President, Research Data Collaborations at Elsevier. Erik Schultes is employed by the GO FAIR International Support and Coordination Office.

### Author Contributions

- Mario Prieto conceived and designed the experiments, performed the experiments, analyzed the data, prepared figures and/or tables, authored or reviewed drafts of the paper, and approved the final draft.
- Helena Deus, Anita de Waard, Erik Schultes and Beatriz García-Jiménez conceived and designed the experiments, analyzed the data, authored or reviewed drafts of the paper, and approved the final draft.
- Mark D. Wilkinson conceived and designed the experiments, performed the experiments, analyzed the data, authored or reviewed drafts of the paper, and approved the final draft.

### Data Availability

All code and Jupyter notebooks, along with raw and processed data, are available on GitHub: https://github.com/Guindillator/Certainty.

### Supplemental Information

Supplemental information for this article can be found online at http://dx.doi.org/10.7717/peerj.8871#supplemental-information.

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
