# Peer review of "Data-driven classification of the certainty of scholarly assertions"

_PeerJ, doi:10.7717/peerj.8871_

## Round 0.1 · original submission · Major Revisions

Both reviewers have made positive comments about your manuscript, which matches my own impressions, although they have also raised some questions that you need to respond to, along with my questions listed below. Reviewer #2 has made some helpful suggestions around wording and extending the literature review, as well as asking some important questions about the design and analyses used. I look forward to seeing your revised manuscript in due course.

Line 32: A kappa statistic might be more useful here as the percentage accuracy depends on the overall distribution of the three levels of certainty.

Line 40: I might be misunderstanding the structure of this sentence, but the comma after “data” here seems spurious if the article contains both the interpretation and reasoning. I’m not sure how the “reasoning behind those conclusions” exists independently of the article, or, in fact, what “those conclusions” means in the absence of previously mentioned conclusions. Based on the above, and assuming I’m not confusing myself here, I think the “continue” on Line 41 relates to the singular “article” on Line 40 and so should be “continues” unless the article and the reasoning are clearly made distinct (if they are distinct, the comma after “conclusions” on Line 41 would appear spurious).

Line 44–46: I wonder if this presents “hedging” too negatively (although I appreciate that the phrasing is itself quite neutral), whereas I would say that, if anything, researchers are often too bold in their claims, perhaps in their desire to publish, attract more funding, etc. (I say this as a biostatistician who often increases the degree of hedging on manuscripts drafted by other researchers). If you agree with this point, could you add a comment that hedging is often appropriate due to concerns about external validity. This could also draw on the “replication crisis”.

Line 51: Spurious “is”?

Lines 60–61: While I completely agree with this point, references to support these statements would be very welcome here.

Lines 79–80: I’m not at all convinced that “degree” implies “discrete/discontinuous”. If anything, I’d be more inclined towards the opposite interpretation and think that degree generally implies a continuous scale, with “type of” or “category of” implying a discrete (nominal or perhaps in some cases ordinal) scale, and with “levels of”, for example, implying a discrete scale more than a continuous one for me. This quote doesn’t seem at all inconsistent with each fragment having a probability (0–100%) of the assertion being valid, for example.

Line 90: If you don’t mind a stylistic suggestion, I’d put “and” before the second “despite” here.

Line 95: Perhaps an “and” prior to the final list item?

Line 117: Spurious space in “manually- curated”.

Line 123: Should this be “…others to USE a numerical scale…” (thank you also to reviewer #2)

Line 129: The responses here would all be ordinal, making ordinary PCA problematic in some respects (through the resulting lack of multivariate normality). Did you use a matrix of polychoric correlations for the PCA? How was the optimal number of components determined (parallel analysis would be my preference here)?

Line 136 (and 380): While I fully appreciate the challenges of publishing a sequence of articles, sufficient information for the reader to judge this model’s validity and limitations is needed in the present manuscript rather than a reference to a manuscript under preparation. It would also be desirable for the code for this to be included in the linked GitHub repository (as per the PeerJ policy on data and code). An alternative would be to have this other publication accepted before returning to the present manuscript. Yet another option would be to remove this component from the present manuscript altogether.

Line 137: Perhaps add “the” after “in” here.

Line 143: Somewhere around here, it would be useful to note that the analyses were performed using Python and give the version number for Python and all key libraries to enable replication.

Line 147 and 156: I’d suggest “of whom” rather than “of which” for people.

Line 177: Spurious space before “%” here.

Line 182: I’d suggest “…indicating no EVIDENCE OF annotator…” rather than the currently absolute statement.

Lines 188–191: This sounds like something from the methods. When it is moved there, it might also be worth emphasising the rank-based nature of Spearman’s correlation coefficient (which has implications in how “weight” is interpreted on Line 189).

Lines 191–203: The HCA isn’t “indicating” the Spearman’s correlations as such, or vice versa. These are different approaches to grouping variables, which will often, but not necessarily if I’m understanding exactly what was done here, produce similar conclusions. Did you consider Kendall’s tau-B which has some advantages over Spearman’s, particularly for data with frequent ties?

Lines 194–202: It would be more conventional to show small p-values as “p<0.001”.

Line 423: While I don’t disagree, “will” here is a very strong prediction to make. It also seems to me that this could be more true for some disciplines than others (reviewer #1 makes a related comment). If you agree with that point, it could also be mentioned around here.

Table 1: Note decimal places are elsewhere indicated with periods and not commas.

Table 5: The decimal places here seem excessive.

Reviewer 1 ·

Basic reporting

The theme of the article is relevant and marks a continuity of research reported in previous articles. Language is clear, professional and unambiguous. Raw data is supplied that enables checking and validating the results.

Experimental design

The article is within the scope of PeerJ, considering this is a research article with focus on rhetoric and methodological issues concerning scientific communication in the biomedical domain.
The research problem is clearly stated. Possible benefits of the research are shown. Methods are adequate and described in detail. The intended audience and beneficiaries are clearly stated, line 405.

Validity of the findings

Maybe the Conclusion, line 441, is too general. As authors themselves suggest, line 51, “As a result, biology papers contain is a wide range of 52 argumentational structures that express varying degrees of confidence or certainty.”
Maybe the Conclusion is valid only within the biomedical domain. No evidence is presented of its general validity or in other domains of knowledge.

Additional comments

Improve the abstract for clarity; on page 6, line 30, Abstract, for example, insert a sentence to make clear the relation of the degree of certainty of scientific claim with text-mining, e.g. , the sentence on page 7, line 65, "This problem is worsened in the context of text mining algorithms that may miss the nuance of a scholarly assertion when extracting the entity-relationships that make up the claim."

Reviewer 2 ·

Basic reporting

The text needs some revisions in terms of English expressions etc. It seems that the submission has been rushed a bit, since there are several typos, repeated text etc that should be corrected.

More specifically:

Line 78: “[e]ach [sentence] fragment conveys a degree of certainty about the" :
I think the use of [] here is erroneous especially in the [e]ach part.

Line 123: "... to or a numerical scale." This is either a typo or an unfinished sentence. Please rephrase

Line 132: "Principle Component Analysis" : I assume you mean Principal Component Analysis but even so, you can use the acronym presented in line 129.

Lines 156-163: This whole part is repeated from lines 147-154

Line 165: Seems as if the beginning of the sentence has been cut off?

Overall, the document would benefit from a thorough revision as currently it seems a bit rushed and hasty and this renders the understanding and appreciation of the presented results difficult.

The literature review is quite limited. The article neglects related work in the field and especially work related to uncertainty on scientific and bio-medical corpora which is the focus of this article.

In terms of older work, some reference to the work of Hyland would be reasonable. And maybe Light et al. (2004) since they also worked on 2 and 3 level uncertainty. Also I would like to see some reference and maybe comparison with the annotations in the BioScope and/or GENIA-MK corpora, or even the CoNLL 2010 shared tasks. How are the selected sentences different if at all to other work?

In terms of user-based surveys:

Malhotra et al. (2013), also uses results of user-based questionnaires

Zerva et al. (2017) also discuss the annotations of uncertainty on a scale of 1-5 among two groups of researchers in biomedicine.

Some discussion and comparison with the conclusions of the aforementioned articles would definitely benefit the paper.

On the conceptual level, other work such as the ones of Szarvas and Vincze is also relevant, and maybe some reference to factuality as well. The work of Soldatova and Rzhetsky might also be of interest in addition to ORCA. Overall the related work is in needs of expansion and more thorough review of recent literature.

In terms of the selection and curation of the 45 statements of the corpus. The source and the selection process for the statements is unclear, and the linked protocol does not seem to have any related guidelines.

In terms of the presentation of the surveys: The actual annotation scale and differences for each survey is not presented clearly in any part of the document and has to be inferred. This is a crucial point for the experiments and result interpretation and should be presented clearly in the beginning of the Materials and Methods section. Also, the statistics of the coverage for each statement of the corpus are not presented very clearly. It was hard to understand what the presented numbers correspond to. More detailed statistics on the coverage for each statement in each survey would be useful.


Indicative references discussed earlier:
Soldatova, L. N., Rzhetsky, A., De Grave, K., & King, R. D. (2013, April). Representation of probabilistic scientific knowledge. In Journal of biomedical semantics (Vol. 4, No. 1, p. S7). BioMed Central.
Light, M., Qiu, X. Y., & Srinivasan, P. (2004). The language of bioscience: Facts, speculations, and statements in between. In HLT-NAACL 2004 Workshop: Linking Biological Literature, Ontologies and Databases.
Malhotra, A., Younesi, E., Gurulingappa, H., & Hofmann-Apitius, M. (2013). ‘HypothesisFinder:’a strategy for the detection of speculative statements in scientific text. PLoS computational biology, 9(7), e1003117.
Zerva, C., Batista-Navarro, R., Day, P., & Ananiadou, S. (2017). Using uncertainty to link and rank evidence from biomedical literature for model curation. Bioinformatics, 33(23), 3784-3792.
Hyland, K. (1996). Writing without conviction? Hedging in science research articles. Applied linguistics, 17(4), 433-454.
Vincze, V., Szarvas, G., Farkas, R., Móra, G., & Csirik, J. (2008). The BioScope corpus: biomedical texts annotated for uncertainty, negation and their scopes. BMC bioinformatics, 9(11), S9.
Szarvas, G., Vincze, V., Farkas, R., Móra, G., & Gurevych, I. (2012). Cross-genre and cross-domain detection of semantic uncertainty. Computational Linguistics, 38(2), 335-367.
Vincze, V. (2015). Uncertainty detection in natural language texts (Doctoral dissertation, szte).

Experimental design

The research question is well defined and interesting.

The rationale for the survey design is reasonable, although I would find the results more convincing if there was better correspondence between the qualitative and quantitative surveys. As it is, S1 and S2 are 4-level and 2-level respectively and both qualitative with different labels for each level/class of uncertainty. S3 is quantitative (i.e., there is no qualitative description of each certainty level/class), but is 3-level. As such, it is rather hard to infer whether the differences observed in the resulting annotations and inter-annotator agreement patterns, are due to the quantitative scale or the 3-level class. This in my opinion somehow weakens the conclusions about the distinguishability of the three underlying certainty categories.

In terms of the clustering it is a bit unclear what was the PCA and clustering applied on. Was each statement clustered using the choice of annotators as data points/features? And the same for PCA? How was the potential difference in the number of annotators per statement handled in that case? Also, if this is the case, in the case of S2 aren't the results limited to a maximum of 4 clusters/components by definition? The Spearman Rank Correlation in Fig. 3 is interesting but rather intuitive and I am not sure it is sufficient to draw conclusions.

Overall, is this the typical analysis statistics for such surveys in the field? I would like some more lengthy explanation of the survey analysis and the rationale for choosing the presented methods and metrics.

Validity of the findings

I would appreciate some more detailed explanation of the rationale for the presented metrics and some more careful discussion of the concluded statements, as I explained in the previous section.

Moreover, I would like to see some qualitative analysis of the findings as well. Based on the examples, the researchers were asked to judge the certainty level of the statements based on the textual information solely. Hence, since it is proposed that there is sufficient evidence supporting a three-level categorisation of uncertainty for scientific statements, it would be very interesting to see some textual analysis of the statements with respect to the attributed categories. Are there some textual patterns supporting the three-level categorisation? Such a finding would further support the proposed application for automated application of the categorisation system.


On a more specific point, in line 357 it is mentioned: "The studies that preceded this one have all suggested four categories of certainty. In this study, we identify only three." This si a very strong statement, and rather unfounded. Which studies specifically have all suggested 4 categories? To the best of my knowledge most studies conclude that multi-class categorisation of uncertainty is rather unfounded (see the experiments of Rubin et al. (2007), Zerva et al. (2017) etc. ). Also most biomedical corpora annotations opt for binary or 3-level annotations of certainty/speculation (see BioNLP ST, Genia-metaknowledge corpus, FactBank corpora, BioScope etc). I have included some references at the end but overall, reconsideration of related studies and categorisation proposals is essential here.


References:
Vincze, V., Szarvas, G., Farkas, R., Móra, G., & Csirik, J. (2008). The BioScope corpus: biomedical texts annotated for uncertainty, negation and their scopes. BMC bioinformatics, 9(11), S9.
Rubin, V. L. (2007, April). Stating with certainty or stating with doubt: Intercoder reliability results for manual annotation of epistemically modalized statements. In Human Language Technologies 2007: The Conference of the North American Chapter of the Association for Computational Linguistics; Companion Volume, Short Papers (pp. 141-144). Association for Computational Linguistics.
Saurí, R., & Pustejovsky, J. (2009). FactBank: a corpus annotated with event factuality. Language resources and evaluation, 43(3), 227.
Zerva, C., Batista-Navarro, R., Day, P., & Ananiadou, S. (2017). Using uncertainty to link and rank evidence from biomedical literature for model curation. Bioinformatics, 33(23), 3784-3792.
Nédellec, C., Bossy, R., Kim, J. D., Kim, J. J., Ohta, T., Pyysalo, S., & Zweigenbaum, P. (2013). Overview of BioNLP shared task 2013. In Proceedings of the BioNLP Shared Task 2013 Workshop (pp. 1-7).
Thompson, P., Nawaz, R., McNaught, J., & Ananiadou, S. (2011). Enriching a biomedical event corpus with meta-knowledge annotation. BMC bioinformatics, 12(1), 393.

Additional comments

I strongly believe that a thorough revision in terms of the use of English, the and the explanation of the surveys and methods would significantly benefit the document.

---

## Round 0.2 · Major Revisions

Thank you for your revisions. As you can see, Reviewer #1 has no further comments. Reviewer #2 has made a number of minor comments that will need your attention. All of their comments should be responded to, along with mine below, on a point-by-point basis where you either explain the changes that have been made in response to that particular point or explain why no changes have been made. In particular, addressing your response around the PCA aspects encouraged me to look more carefully at some of the other analyses and your code for these, and this has raised some new questions. I apologise for not identifying these with the initial revision. If I am misunderstanding some aspect of the data analyses, I’m very happy to be enlightened/corrected here, but I’d hope that I wouldn’t be the only reader to misunderstand in that case and so additional explanations in the manuscript might still be indicated.

While I can see your interpretation of the guidelines around p-values (Lines 398–407), I have always seen this as intended to prevent dichotomisation into non-significant/significant. As an author, reviewer, and editor for PeerJ, I use/see p-values with “<0.001” for small p-values. Given that p-values become, in relative terms, unreliable below some level for small changes to individual data points in smaller data sets and/or departures from the model assumptions, I think this is a sensible approach and one that you are welcome to use.

As a trivial but related point, for percentages where the denominator is less than 100 (e.g. Line 458), I don’t see the value in providing decimal places since each observation (statement here) contributes over 1% in itself.

Thank you for clarifying the approach used for PCA here. However, as this is performed separately for each survey, this appears to be based on something very close to compositional data (if you know the number responding “s2-Relatively High” for a given statement, you also know [approximately, due to slightly different numbers answering for each statement] the number responding “s2-Relatively Low”). Note that the standard deviations in that case would be the same for both levels in survey 2 and so the normalisation would not change this relationship. For survey 1, this property of the data would mean that if the number answering each statement was identical, the 4th eigenvalue would always be 0 and only 3 eigenvectors could ever be obtained. (The same would apply for the other surveys with a constant number of respondents, the last eigenvalue would be 0, producing at most one fewer component than the number of levels.) This is the motivation for compositional approaches to PCA and why ordinary PCA cannot be used for compositional data. The 97.4% of variance explained by the first 3 factors (Lines 427–428 and Table 6) using the actual data reflects that small amount of noise in terms of the number answering for each statement rather than the structure of the data demonstrating only three and not four factors. The pattern of positive and negative loadings (Lines 515–516) is also an expected result of this compositional structure to the data (with perturbations from the different numbers of respondents per statement). While you could change the standardised counts to percentages and use compositional PCA (there are several packages for this available on CRAN), this could only ever identify three components with the 4th percentage being a linear combination of the others. This would suggest that with more options offered (providing a finer granularity of certainty), more components could have been identified.

This particular set analyses could instead be performed at the individual level, perhaps using item response modelling. Even a simple Rasch model (providing an item threshold score for each response above the lowest and a person propensity score on the same scale for each participant) would be useful in seeing whether the item thresholds are all distinct or if, for example, two are very close together, which would suggest that they could be merged. If all three cutpoints were distinctly separated, and this can be compared to the distribution of participant scores to see if the differences are meaningful there, this would support at least four levels of certainty; if two were close together, this would support three; etc. If I am still misunderstanding these analyses, my sincere apologies!

Some of the Spearman’s correlations in Figure 3 are similarly affected (overall, there has to be a strong negative association between the two survey 2 responses, rho=-0.97 here and this would, by definition, be -1 if the number responding to each statement was constant). This is slightly more subtle but still evident for surveys 1 and 3. You could supress the within survey correlations to avoid readers over-interpreting these. You’ll note that most within-survey correlations for distinct levels are negatively correlated as expected.

The same property would encourage different response levels to be in different clusters using clustering of responses. There are cluster methods for compositional data in robCompositions, for example, that address this dependence within the data.

I don’t know if it’s completely clear to the reader that the NbClust results (Lines 322–332) are based on the statements rather than the classifications which have been the focus until this point. The discussion on this (Lines 499–510 and 543–544) could also make this point a little clearer.

Is the mean of 16 responses per participant for survey 2 (Line 365) correct? Based on 3000 responses in total, for 215 participants, I get n=14 for the mean.

I should perhaps note that I think the overall weight of evidence is towards three levels of certainty here (the statement clustering is reasonably convincing for me and the reliability analyses provide support for this), but the PCA and correlational analyses do not provide the same level of support here as the other approaches if I am understanding these analyses correctly.

Reviewer 1 ·

Basic reporting

This version of the paper is now more robust and concise. The questions relating to the clarity and precision of the conclusions we have raised were considered

Experimental design

The methods are described in detail.

Validity of the findings

The paper clearly demostrates the validity of the findings

Reviewer 2 ·

Basic reporting

The text and use of English have been greatly improved.

I have some minor observations:

lines 33-35: The terms author-annotated corpus and publicly-annotated corpus initially confused me. Since you are using this distinction (or any other term you decide), it would be good to mention it in the main text as well, and use it consistently, so that we always know which of the two corpora you are referring to.

lines 158-161: It would be better to specify in which of the aforementioned publications this is the case, otherwise the statement is a bit dubious unless it holds for all annotations.

line 172: I am a bit confused with the term "fidelity". does it mean consistency, or something else?

lines 209-211. Perhaps here is a good point to clarify whether this is the "author-annotated" or "public-annotated" corpus, introduce the naming convention properly and stick to that name henceforth.

lines 291-292: I would avoid the repetition of the word "represent" in this sentence, it distorts the meaning. Perhaps the first instance could be replaced by "constitute/are/" or "depict/exhibit/illustrate" (I think the former are closer to the intended meaning)

line 295: Spearman’s formula rank --> Spearman’s formula ranks

lines 438-439: Perhaps "...was used to build a neural network model" should be replaced with "...was used to train a 5 layer NN model"

line 441: Is there a comma needed after "Survey 3"?

lines 464-465: after the comma, the sentence does not make sense. what does "this in these " refer to? and "more detailed arguments"? Is there a verb missing? Also, please clarify which corpus you refer to since you have mentioned two versions.

line 506-507: It would be better to use consistently either numbers or words to represent numbers especially if referring to the same thing (11 indices versus six indices)

General comments:
Shouldn't equations be numbered?
There is inconsistent use of quotation marks when quoting other authors (e.g., compare line 277 to line 147.

Experimental design

I am satisfied with the explanations, evidence, and support for the experimental design. Just some observations:

lines 195-227: In the light of the new text added, I am wondering about the reasons/motivation the TAC dataset was chosen (as opposed to a collection of abstract for example). Since TAC was generated for citation identification and summarisation, did you use any such information to select the sentences?

lines 346-347: The typical reporting of performance results when it comes to Precision/Recall typically includes F-score, unless for some specific reason this cannot be calculated. But in this case, I do not understand why F-scores are not presented, especially since that sentence is supported by some provided references, which, as far as I could see, do use the typical Precision-Recall-Fscore schema. F-score can be calculated from the Precision/Recall scores you provide, so I would like to see it in the tables and evaluation methods.
That said, I am not sure why these references are provided at that part of the text. In my understanding, there is no direct/indirect comparison to their results. Are they cited just because they used a similar evaluation schema? I think the role of this citation could be clarified (or the citation removed if they do not contribute something).


Line 438: How was this corpus generated? I am afraid I am a bit confused with the two versions of the corpus. Aso, please stick to the same names used.

Validity of the findings

Lines 410-413: Is this X indices out of 30 that were available in NbClist? Please specify this.

Table 7: Please add F-scores, so that the results are complete and aligned with the standardised way of presenting machine learning results in such NLP/text mining tasks. Also, it would be nice to have the confusion matrices as well, if there are no space concerns.

I understand the limitation of the authors in terms of annotations for the surveys and sympathise with the difficulties involved with carrying out lengthy annotation surveys". I appreciate their response to my previous question and the explanation on the quantitative/qualitative nature of S3. My point is that while I agree on the potential bias introduced by the "named" labels as opposed to the enumerated ones, in order to properly examine/remove the bias, we would need either exclusively enumerated labels, or an experiment with enumerated/named labels on the same number of classes. E.g.,: Sx = {1,2,3,4} and Sy = {high, medium-high, medium-low, low}.
However, since the relevant claims were softened and the limitations explained in detail, I do not believe that this discrepancy should interfere with the acceptance of this paper, but perhaps the issue could be addressed in future work. With the explanation in lines 529-543, the reader gets a very clear view of the experiments, findings and related implications.

---

## Round 0.3 · Minor Revisions

Thank you for your revised manuscript. The one remaining reviewer pointed out a couple of typos, but otherwise was satisfied with your revisions. I am still of the view that your finding of three components/clusters is potentially due to not looking at more than four levels of certainty (see Table 6, where three components explains all of the variance given the compositional nature of the data) and while I don’t personally expect that this would change your findings, it is one thing that could be investigated in future research on this topic.

I’ve made some other comments below for you to also consider when revising your manuscript. Some of these are editing suggestions as I think we are getting close to the point where your manuscript could be accepted (subject to addressing these comments).

Line 218: The word “transformations” seems to be missing after “centered log-ratio” here. (As you do on Line 307, although I’d prepend “the” or make “transformation” plural there.)

Line 273: For consistency and clarity, you could insert “S2 and S3” here, resulting in: “…(S1) or Qualtrics (‘Qualtrics’, S2 and S3)…”

Line 296: Apologies for not spotting this last time, but I think you mean “Pc” here and not “G” (for all three instances on this line). The probably of chance agreement (assuming homogeneity of categories) with 4 categories is 4 x (1/4)^2=0.25 (or simply 1/4=0.25 after conditioning on the first rater); similar for 2 and 3 categories. I think the same point applies to Lines 514–515 where Pc=0.5. Note also the slightly inconsistent spaces around equals signs (Line 296, none for 2 instances; Line 296, one with a trailing space; Line 299, three with spaces on both sides) and this should be checked throughout the manuscript.

Line 307: See above.

Line 309: “go” here seems spurious (or did you mean “g(x)” to indicate the geometric mean?) and “after” should, I think, be “before”—this is the log of the ratio. To quote the reference you used, Reimann, et al., “a centred logratio (clr)-transformation (Aitchison, 1986) where, in order to construct the logratios, each variable is divided by the geometric mean of all elements measured, followed by a log-transformation”)

Line 330: I’m not quite sure what “Normalization allows the comparison of two nominal variables on different scales.” is intended to mean here. Nominal scales are categorical without ordering. All of the certainty scales are at least ordinal (categorical with ordering) and the counts of responders are on the absolute scale.

Lines 330–345: I found parts of this paragraph to be ambiguous and it seems to be in need of some rewriting and/or restructuring. I think readers will be helped if you mention that you’re calculating and then using, I assume, the clr transformed values (as described above) here. If this is all being done on the transformed variables, then I’m not sure what “normalized for the different number of annotators” would mean (for compositional data, the number of annotators is no longer relevant) and this (Lines 331–332) is not otherwise linking back to the clr transformation. However, given the reference to the contingency table (Line 339), it looks as if the PCA was being performed on the compositional (normalised) counts, and so compositional PCA would be needed (as stated on Line 304, and I’d always make it clear that compositional PCA techniques are being used when this is the case). There also seems to be a small conflict between Line 336’s reference to clustering on the PCA data (which made me think you were doing this on the component scores) and the reference to contingency tables being the input to both procedures (Line 338). I appreciate that you provide the code in Python, which would answer all of my questions, but the methods also need to be clearly stated here. Perhaps a flow diagram (around Line 302) would be a clear way to set out your data processing and analyses before you explain these in text?

Line 364: Is “We report that…” hedging? For example: “We report that swans are always white.” is a statement of absolute certainty (requiring only a single counter-example to falsify).

Line 398: 7/45 would be 15.555%, rounded to 16% as an integer percentage (not 15%). See also Line 406.

Line 414: 3/34=8.888 or 9% (not 8%).

Line 468: Some readers will want the meaning of the value after the ± to be explicitly stated here.

Line 575: “moderateLY”?

Line 578: “Zero”?

Figure 3: The legend should make it clear that this is based on (I assume) the clr transformed values.

Table 3: The decimal places for the percentages varies here with most being one (e.g. 2.2%) but some two (e.g. 11.11%) and some none (e.g. 20%). See also Table 4 for 13.32% and Table 5 for 11.11%, 22.22%, and 31.11%.

Table 6: This makes it clear that you couldn’t detect four levels as there are only three independent levels (the cumulative proportion of the variance is 1.000 with three components).

Reviewer 2 ·

Basic reporting

No major comments, I am satisfied with the response of the authors. Just two typos:

-lines 375 and 376: Extra parentheses in the Recall and F-score formulas
-line 473: Full-stop at the end of the sentence missing

Experimental design

No more comments, I am satisfied with the response of the authors.

Validity of the findings

No more comments, I am satisfied with the response of the authors.

---

## Round 0.4 · Minor Revisions

Thank you for your revisions, which I think address almost all of my comments and all of those of the remaining reviewer sufficiently well. There are a reasonable number of minor editing comments, more than I think it is wise to leave to the proofing stage, and I still have some questions about the interpretation of three versus three or more categories with my comment that “I am still of the view that your finding of three components/clusters is potentially due to not looking at more than four levels of certainty (see Table 6, where three components explains all of the variance given the compositional nature of the data)” still needing some further attention/rebuttal. You said that “It would be very interesting to know if adding more categories changes the outcomes, though we don’t expect this will be the case.” but I wasn’t able to see your reasoning for this conclusion in the rebuttal or manuscript and I think that some readers will have the same question as I do. I will ask for your patience with another round of revisions.

It would be great to see the final version of all the notebooks on GitHub before accepting your manuscript and some of my questions below might have been able to be answered by looking at the code and results there. NbClust.ipynb in particular seems to be out of date (currently consisting of the analyses and results prior to your switch to compositional analysis). My apologies if I’m missing the current versions somehow.

Line 47: Given that you have used compositional analyses, do you mean “there are [at least] three” here as without a survey using 5 categories, I still can’t see how you’ve ruled out 4 categories for your compositional data analyses. Also Lines 343–344, 528, 538, 596, 608, 695 (and elsewhere). You might wish to describe the different sources of support more distinctly (e.g. the inter-rater reliability analyses versus the compositional PCA) to make it clear what you can conclude from each source.

Lines 76, 455, and elsewhere. My apologies for the pedantry, but you seem to switch between “Figure” and “Fig.” in the text without, as far as I can tell, a systematic rule.

An extremely trivial point, there is a missing period after “et al” on Line 116.

As another trivial suggestion, Line 119’s “no certainty, low, moderate, and high certainty,” could be simplified (by a word) to “no, low, moderate, and high certainty,” as “certainty” is the noun for all four adjectives here (there doesn’t seem to be anything special about “no” to me).

Line 152 lists categories of “strong, moderate, weak”, followed by “definitive”. This seem to be the only time you don’t order categories in a monotonic fashion (setting aside the “cannot evaluate” category, Line 119, which isn’t necessarily part of an ordinal measure here, as you allude to on Lines 620–623). Note that when listing categories in the introduction and elsewhere, you sometimes switch between low->high and high->low orderings.

Line 227: Perhaps add an article (definite here) to “using [the] k-means algorithm”?

Line 231: And perhaps delete the (indefinite) article from “To evaluate its accuracy, [a] 20-fold Cross-Validation (CV) was used.”

Line 261: Perhaps “participants” (as you use elsewhere, including this paragraph) rather than “subjects” here?

Line 297: Should the “G” here also be “Pc”, as on the line two below? Or is the “The formula for G is defined…using…” See also Line 488.

Line 301: I think you are missing a “0–“ before the “0.2” here. I’d also suggest “0.00”, “0.20” and “0.40” on this line to match the “0.60”, “0.80” and “1.00” on the following line (i.e. 2 dp for all rather than a mixture).

Line 328: The intended meaning of “All Spearman interactions are based on hypothesis testing.” isn't clear to me. Do you mean “All Spearman associations are interpreted based on hypothesis testing.”, or “….correlations…” if you prefer, or something else here?

Lines 367–368: I think you either need an article “…[a] 20-fold CV scheme…” or you could simply delete “scheme” instead.

Line 401: Should this read “Medium Low [was]” (rather than “were”)?

Line 416: Just “…of the total…” (not “totals”)?

Line 417: You use “sentences” here where you otherwise (including in this paragraph) use “statements” with the exceptions of Figure 1’s legend and Line 137 being the only other occurrences I could find for “sentences” in this context.

Line 419: I don’t think you need “rank” after “Spearman correlation” here (“Spearman rank correlation” would also be common), unless I’m missing your intention of course.

Line 432: Missing space in “contains theS1-Low”.

Line 436: Spurious capital (“L”) in “NbCLust”.

Line 446: I’m not sure that “row 1”, etc. will be immediately obvious to the reader since these refer to the physical rows number 7, etc. in the table. I’m also not sure why you’re drawing the reader’s attention to the SD when the variance is on row 0/6 and is the measure used for the other two reported here. Do you need the SD in the table?

Lines 454 and 542–543: I believe that this is a mathematical result rather than an empirical one and, assuming I am correct, it seems that the reader might not fully understand the source of this result (i.e., the model and not the data). With the compositional analysis approach, will you not always be limited to k-1 components at most because k-1 components will always explain all of the variability in the data (and with the kth component being mathematically fully determined by these k-1 components, it cannot provide additional information)?

Line 460: “Low [has] a” (rather than “have”).

Line 467: You could delete the article here: “…using [a] 20-fold CV…” (or add “scheme” as used on Line 368).

For my previous comment: “Line 468: Some readers will want the meaning of the value after the ± to be explicitly stated here.”, while I appreciate that you’ve addressed this on Line 468 with “indicating that it achieved 89.26% accuracy with +/- 2.14% of standard deviation.”, I wonder if this would be slightly clearer as “indicating that it achieved 89.26% accuracy with a standard deviation of 2.14% between folds.” (I think this is what you have calculated based on the code and output in your notebook for the cross-validation results.)

Line 501: This (“with statistical significance, because there were only two choices”) seems to have the comma in the wrong place, before “because” rather than before “with”, but also the p-value is based on a chance-adjusted statistics (reflecting the number of categories rather than the marginal distribution as per kappa, but still chance adjusted) so the logic doesn’t follow as I’m reading what is currently written here. With fewer categories, there will be more agreement by chance (as you say on Line 516), but the test statistic incorporates this information and so the p-values remain unaffected. You could simply delete this fragment, I think.

Line 513: I don’t think you need this comma (before “than”).

Line 574: Perhaps “using Spearman correlations.” (removing “a”).

Line 576: Do you mean “…are also moderately [to] highly correlated” here?

Line 663: Looking at the reference on Line 857, you give the authors as “Saggion H, Ronzano F, Others.” but according to the page itself and the PDF linked to on https://www.aclweb.org/anthology/W16-1520/, they were “Horacio Saggion, Ahmed AbuRa'ed, Francesco Ronzano” (in this order, and with three authors, which you would reference using all three names elsewhere, e.g. Line 746). Am I looking at the wrong publication here or is there an issue with this reference? If the latter, it might be worth checking other publications with “Others” included in the author list.

Figure 3: The decimal places are slightly inconsistent here (e.g. “0.8” and “0.72”). Note that you have this corrected on Lines 425 and 431, for example.

Table 6: I can’t see how you have loadings for 4 components here if this is from a compositional PCA with 4 categories. Am I misunderstanding something here, perhaps some adjustment in the functions you are using to resolve, inappropriately here I would think, the collinearity? The 4th component explains none of variance (as everything is explained by the other 3 components) and so shouldn’t be estimated if I understand correctly. As noted above, I couldn’t find the updated code on GitHub to check my understanding of what was done here.

---

## Round 0.5 · accepted · Accept

Thank you very much for your thorough and thoughtful responses, yet again, in particular to what I see as challenging issues in the data analyses here. I’m still of the view that ruling out higher numbers of categories cannot be confidently achieved without more categories in the collected data (including for statements, where this would seem to depend on, for example, “true” categories existing in between the provided categories, with some respondents going one way and others the other way—the relative positioning of “true” and provided categories influencing whether or not statistically significant evidence could be found for > n clusters of statements in the simulations I’ve looked at), but this perspective certainly shouldn’t hold up what I regard as an interesting and important publication. There might also need to be further work on the statistical methods for such studies, based on yet more simulations to show how well true models can be recovered.

Overall, I suspect that your conclusions are likely to be true and your work has certainly highlighted some important questions going forward, as good research should. It is reassuring that your interpretations and conclusions remain robust with alternative analyses and I appreciate your patience and diligence in looking so carefully at these. I am delighted to accept your manuscript as what I regard to be a very useful piece of the puzzle for this important topic.

As a small point for you to address in the proofing process, I think the citation “(Chollet and ogguithers, 2015)” (Line 371) needs attention. See the suggested citation on https://keras.io/getting-started/faq/#how-should-i-cite-keras, namely (in BibTeX):

@misc{chollet2015keras, title={Keras}, author={Chollet, Fran\c{c}ois and others}, year={2015}, howpublished={\url{https://keras.io}},}

There are also some inconsistencies throughout the references (e.g. non-italicised journal names, such as on Lines 728–729 and 734; missing volume/issue/page numbers, such as on Line 734; inconsistent source title capitalisation, compare Lines 734 and 782 with Lines 742 and 791–792; and seemingly inconsistent abbreviations, such as on Line 790 compared to Line 793). This is not necessarily an exhaustive list of lines for each point and each reference should be carefully checked for all essential details being provided and for consistency.

Finally, the comma in “The intermediate values Medium High (S1) and Category 2 (S3), are found…” (the comma is on Line 576) does not seem necessary to me.

Congratulations again on your work. I look forward to seeing the discussion it will generate, both amongst researchers specialising in this area and all researchers who have to describe their own or interpret the reporting of other’s study findings.